# Uncovering the SUMOylation and ubiquitylation crosstalk in human cells using sequential peptide immunopurification

Frédéric Lamoliatte[1,2,*], Francis P. McManus[1,*], Ghizlane Maarifi[3], Mounira K. Chelbi-Alix[3] & Pierre Thibault[1,2,4]

Crosstalk between the SUMO and ubiquitin pathways has recently been reported. However, no approach currently exists to determine the interrelationship between these modifications. Here, we report an optimized immunoaffinity method that permits the study of both protein ubiquitylation and SUMOylation from a single sample. This method enables the unprecedented identification of 10,388 SUMO sites in HEK293 cells. The sequential use of SUMO and ubiquitin remnant immunoaffinity purification facilitates the dynamic profiling of SUMOylated and ubiquitylated proteins in HEK293 cells treated with the proteasome inhibitor MG132. Quantitative proteomic analyses reveals crosstalk between substrates that control protein degradation, and highlights co-regulation of SUMOylation and ubiquitylation levels on deubiquitinase enzymes and the SUMOylation of proteasome subunits. The SUMOylation of the proteasome affects its recruitment to promyelocytic leukemia protein (PML) nuclear bodies, and PML lacking the SUMO interacting motif fails to colocalize with SUMOylated proteasome further demonstrating that this motif is required for PML catabolism.

[1] Institute for Research in Immunology and Cancer, Université de Montréal, P.O. Box 6128, Station, Centre-ville, Montréal, Québec, Canada H3C 3J7. [2] Department of Chemistry, Université de Montréal, P.O. Box 6128, Station, Centre-ville, Montréal, Québec, Canada H3C 3J7. [3] INSERM UMR-S1124, Université Paris Descartes, 75006 Paris, France. [4] Department of Biochemistry, Université de Montréal, P.O. Box 6128, Station, Centre-ville, Montréal, Québec, Canada H3C 3J7. * These authors contributed equally to this work. Correspondence and requests for materials should be addressed to M.K.C.-A. (email: mounira.chelbi-alix@parisdescartes.fr) or to P.T. (email: pierre.thibault@umontreal.ca).

Protein SUMOylation corresponds to the reversible conjugation of small ubiquitin related modifier (SUMO) on the side chain amine group of a lysine residue on a target protein. SUMO plays essential roles in protein translocation, DNA damage response and cell cycle progression[1–6]. Like other ubiquitin-like (UBL) modifiers, SUMOylation involves a cascade of three enzymes: the E1-activating complex SAE1/SAE2, the E2-conjugating enzyme UBC9 and one of the several E3 ligases (such as PIAS superfamily or RANBP2)[4,6]. SUMO maturation and deSUMOylation are carried out by Sentrin SUMO specific proteases (SENP). SUMO was first known to modify its canonical consensus sequence ψKxE/D (where ψ is an aliphatic residue and x any amino acid), however numerous studies reported other consensus sequences such as a phospho-dependent sequence, reverse consensus and non-consensus regions[7–9]. In human, three paralogs of SUMO are expressed ubiquitously (SUMO1, 2 and 3) in all cells, while SUMO4 is expressed in specific organs (kidney, lymph node and spleen), and SUMO5 was recently reported to be expressed in testes and blood cells[10]. Previous reports indicated that SUMO can interact with ubiquitin in a synergic or an antagonist manner[1,11–13]. Moreover, mixed chains of SUMO and ubiquitin have been identified in different studies, although their functions remain unknown[14,15].

The identification of endogenous SUMOylation sites by mass spectrometry (MS) remains a challenge due the highly dynamic nature of SUMOylation, and the complex MS/MS spectra arising from the branched SUMO remnant of tryptic peptides. To overcome these problems, we previously generated a 6xHis-SUMO3-Q87R/Q88N mutant that facilitates the identification of SUMOylated peptides by MS[16]. This mutant releases a five amino acid SUMO remnant that can be immunoprecipitated using an antibody to enrich for SUMO-modified peptides[17]. Similar approaches such as the SUMO3 T90K mutant[18] or the SUMO2 T91R that conveniently use the commercially available anti-di-glycine antibody have been previously developed for the identification of SUMO sites[19]. Moreover, SUMO mutants for which all lysine residues are replaced by arginine residues were used to allow for nickel-nitrilotriacetic acid (NiNTA) purification after Lys-C digestion[20]. More recently, the combination of lysine labelling with the overexpression of a wild-type (WT) like mutant has been reported[21]. While these approaches have been designed to enrich SUMOylated peptides from complex cell extracts, they cannot be used alone to uncover the prevalence and significance of crosstalk between UBL modifiers. To address this limitation, we developed a combined immunoaffinity enrichment strategy that enables the identification of UBL-modified proteins and applied this method to examine crosstalk between SUMOylation and ubiquitylation in the context of protein degradation. Using this approach, we found several interplay between SUMO and ubiquitin including the co-regulation of SUMOylation and Ubiquitylation levels on deubiquitinase enzymes and the SUMOylation of the proteasome for its recruitment to promyelocytic leukemia protein (PML) nuclear bodies (NBs).

## Results

**Optimization of SUMO peptide immunoaffinity purification.** The strategy to identify SUMOylation sites in human cells relies on our previously designed SUMO mutant (Fig. 1a). To improve the method we optimized both the immunopurification approach and the MS analysis of SUMOylated peptides (Fig. 1b). Cells stably expressing the 6xHis-SUMO3-Q87R/Q88N mutant (HEK293-SUMO3m) produce a functional SUMO3 cleavable by trypsin near its C-terminus. After protein extraction from whole cells, SUMOylated proteins are enriched on NiNTA

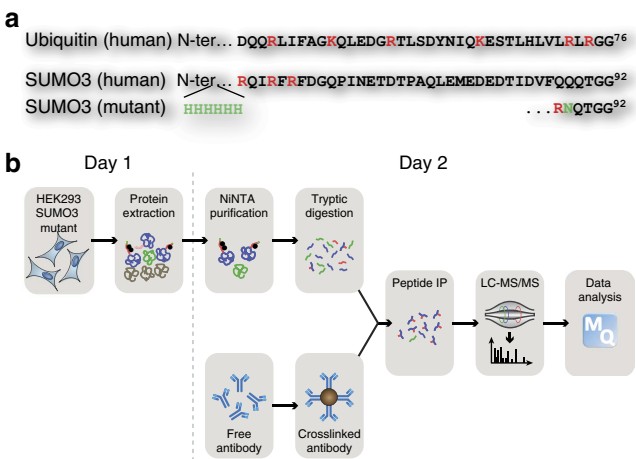

**Figure 1 | Optimization of a SUMO remnant immunoaffinity purification strategy.** (**a**) Protein sequences of the endogenous ubiquitin, endogenous SUMO3 and SUMO3m. (**b**) Overview of the remnant immunoaffinity purification. Cell lysates are subjected to a NiNTA column to enrich SUMOylated proteins before tryptic digestion. Peptides containing the SUMO3m remnant are enriched using the anti-K-(NQTGG) antibody. Subsequent peptides are injected on a Tribrid Fusion. Peptide identification is performed using MaxQuant.

column before their digestion on beads. Desalted and dried samples are reconstituted in an immunopurification incubation buffer. Tryptic peptides that contain the SUMO remnants are enriched using an anti-K-(NQTGG) antibody.

We first developed a high sensitivity MS method on the Orbitrap Fusion, where the automatic gain control (AGC) is used to control ion accumulation rather than varying the injection time. Previous studies have shown that an increased AGC setting led to higher number of identification and a decrease in the total number of MS/MS spectra acquired in a single run[18,22]. Accordingly, we used a fixed AGC value with long injection times to reach the target AGC value while keeping the injection time below 3 s (Supplementary Fig. 1). We found that an AGC of 5,000 yielded similar number of identification and score compared with higher AGC values and significantly reduced the injection time to maximize MS utilization. We further analysed the benefits of this sensitive method with biological samples where we obtained $375 \pm 24$ sites for the developed method (AGC of 5e3 and a 3,000 ms injection time) while garnering only $110 \pm 15$ site using the classic method (AGC of 1e5 and a 50 ms injection time) for the MS/MS scans.

To enrich SUMOylated peptides, we previously used an anti-K-(NQTGG) antibody and performed in solution purification with further washing and elution steps on 30 kDa centricon membranes. This method enabled the enrichment of SUMO peptides from tryptic digests to a level of $\sim 12\%$ (ref. 17). To improve the recovery yield of SUMO peptides, we cross linked the antibody to protein A/G magnetic beads, and optimized the binding conditions including the loading capacity (Supplementary Fig. 2). The cross-linking of the antibody to magnetic beads with dimethyl pimelimidate (DMP) prevented the release of heavy and light chain fragments on low pH elution. Optimal binding conditions were obtained for 2 μg of antibody per μl of bead volume with a crosslinker concentration of 5 mM of DMP. We subsequently determined the optimal antibody:ligand ratio to deplete the pool of SUMOylated peptides from a tryptic digest of 2 mg of NiNTA purified proteins

from MG132-treated cells. We compared the recoveries of SUMOylated peptides of the cross-linked antibody approach with that of our previously published in solution binding strategy (Supplementary Fig. 3a, Supplementary Data 1). While both methods gave optimal results for the same antibody:ligand ratio (1 mg of antibody for 2 mg of NiNTA protein digest), the cross-linking approach led to higher peptide recoveries ($568 \pm 6$ SUMO peptides versus $235 \pm 39$ SUMO peptides) and enrichment levels ($34.7 \pm 0.4\%$ versus $4.5 \pm 0.7\%$) compared with the in solution purification. We also determined the number of SUMOylated peptides identified by MS when using the in solution and the cross-linked immunopurification methods, and found that the cross-linked antibody approach provided an 85% overlap of SUMO peptides using the in solution method, and yielded a 2-fold gain in new identifications (Supplementary Fig. 3b, Supplementary Data 1). To further optimize the method, we performed a titration, where we compared the number of identified SUMOylated peptides with increasing amount of NiNTA protein digest. Using a single LC-MS/MS run, the number of identified SUMOylated peptides reached a plateau for immunopurified samples exceeding 4 mg of NiNTA protein digest, where we obtained 1,046 SUMOylated peptides corresponding to an enrichment level of 50.7% (Supplementary Fig. 3c, Supplementary Data 1). We noted that the proportion of non-SUMOylated peptides increased significantly beyond 4 mg of protein digest, possibly through the displacement of hydrophilic SUMOylated peptides from the reverse phase pre-column by hydrophobic tryptic peptides. Closer examination of the immunopurified extract revealed that SUMOylated peptides are more hydrophilic with a lower proportion of aromatic and aliphatic residues compared with their non-modified counterparts (Supplementary Fig. 4). Lastly, we evaluated different incubation buffers to improve the recovery and the enrichment level of SUMOylated peptides (Supplementary Fig. 3d, Supplementary Data 1). We observed that an incubation buffer containing 0.1% NP40 lead to higher number of identification with an average of 1,277 SUMO peptides and higher enrichment level (81.1%). However, traces of NP40 were still detected in the LC-MS/MS analysis, which compromised the identification of SUMOylated peptides over repeated injections. Accordingly, we opted to use phosphate buffered saline (PBS) containing 50% glycerol as the buffer of choice, which led to the identification of 1,118 SUMO peptides with an enrichment level of 62.9%.

We benchmarked our approach by profiling the changes in protein SUMOylation on treatment of HEK293 cells with 10 µM MG132, a proteasome inhibitor known to affect the levels of protein ubiquitylation and SUMOylation. We evaluated the effect of MG132 over an incubation period of 8 h to further compare the improvement of the present protocol with that of our previous study[17]. It is noteworthy that cell viability for either HEK293 or HEK293-SUMO3m cells remained unaffected over the course of these experiments. We evaluated the reproducibility using three biological replicates originating from different cell cultures with three technical replicates each, where protein extracts were separated and the workflows performed in parallel. A total of 1,640 SUMO sites were identified on 1,983 SUMOylated peptides across all replicates using 4 mg of protein extract per replicate. Among all identified SUMOylated sites, we found that 72% were shared across the different biological replicates (Supplementary Fig. 5). Moreover, reproducible label-free quantification was obtained for all biological replicates with a Pearson coefficient >0.9.

We also noted that an increasing number of new peptide identifications were obtained with each additional replicate (Supplementary Fig. 5). Indeed, we found that a second and third injection provided a gain of 21 and 7% in new identification, respectively. We surmised that the stochastic nature of MS/MS acquisition and the low abundance of SUMO peptide ions might partly explain this observation. This is consistent with the gain in identification associated with the increase in MS2 injection time, suggesting that SUMO peptides of low abundance required better ion statistics for proper assignment.

**Comprehensive SUMO proteome analysis.** As described previously, a limited number of identification was obtained from a single LC-MS/MS analysis of immunopurified samples for an equivalent of 4 mg of protein extracts. To increase SUMO proteome coverage and minimize sample overloading on the reverse phase pre-column for injections exceeding 4 mg of digest from NiNTA purified proteins, we fractionated the samples using strong cation exchange (SCX) spin tips before LC-MS/MS analysis. Two technical replicates were prepared using 16 mg of protein extract from MG132-treated cells, and SUMOylated peptides were purified as indicated above. After immunopurification, peptides were separated into 6 SCX fractions, each replicate corresponding to 8 mg of cell extract (~40 million cells). This approach yielded an unprecedented number of identification with 9,816 SUMO sites on 3,405 proteins corresponding to enrichment levels above 70% (Supplementary Data 1 and 2). In contrast, the same analysis performed using reverse phase LC-MS/MS only yielded 1,170 SUMO sites (Supplementary Fig. 3c). It is noteworthy, that 2D-LC-MS/MS experiments performed using the in solution antibody binding protocol previously identified only 954 SUMO sites[17]. The further enrichment of SUMOylated peptides observed here is partly explained by the sample decomplexification which enabled the identification of lower abundance SUMOylated peptides, and the increased capacity of the SCX column that facilitates the retention of hydrophilic SUMOylated peptides that would otherwise be displaced on the reverse phase trapping column by other tryptic peptides (Supplementary Fig. 4).

To determine the depth and comprehensiveness of the present SUMO immunoaffinity enrichment approach, we compared our data with those of recent large-scale SUMO proteome studies (Supplementary Fig. 6)[17–21,23–28]. Among the 6,288 SUMO sites reported in 11 large-scale studies using different treatments and cell types, 47% of these sites were identified in our study while the remainder was detected mostly in single studies[29]. We observed that SUMOylated peptides of higher intensity are most consistently observed across different studies suggesting that highly abundant SUMOylated peptides are more likely to be identified when using different cell types and stimuli.

To gain further insights into the nature of SUMO peptides identified in the present study, we analysed the sequence motifs of amino acids surrounding the SUMOylated lysine residues (±6 amino acids) using Motif X (ref. 30), and identified 35 motifs clustered into 10 different groups (Fig. 2a). We determined that 12% of all the identified sites resided in the full consensus motif ψKxE/D and 13% in the partial consensus motif xKxE/D. In addition to the previously known consensus motifs, new motifs emerged in this study, including the partial aliphatic and aromatic sequences ψ/φK (14%) as well as the corresponding inverted sequences Kψ/φ (13%), where φ represents Phe or Tyr residues. Moreover, another group of consensus motifs showed characteristics arising from the canonical consensus and the inverted sequences: DIK and KLE. Finally, 25% of all the sites were not attributed to a consensus sequence. To understand the unusually low proportion

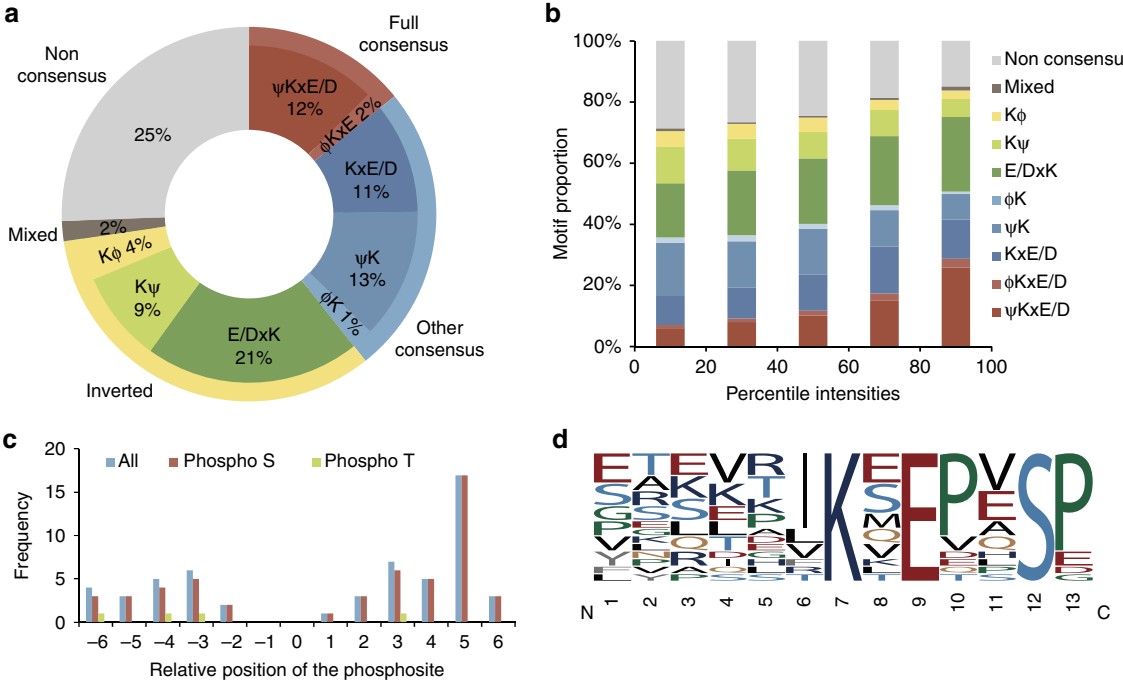

**Figure 2 | Motif analysis of SUMO-modified lysine residues.** (**a**) Pie chart distribution of identified SUMO3 sites based on sequence motif. SUMO3 sites located within a full consensus sequence are represented in red, partial in blue, inverted in green and non-consensus in grey. (**b**) Distribution of the identified SUMO motifs with regards to the intensity of their respective peptides. High abundance peptides are mostly represented by full consensus sequences while low abundance peptides show partial or non-canonical consensus sequences. (**c**) Distribution of the phosphorylated residues with respect to the SUMOylation sites identified on a peptide. (**d**) Phospho-dependent motif identified by Motif X using all the phosphorylated SUMOylated peptides.

of consensus sequences observed in the 2D-LC experiments, we mapped the distribution of motifs according to peptide intensities (Fig. 2b). This distribution revealed that common consensus motifs such as xKxE/D or E/DxKx are over represented in high intensity peptides, while other non-consensus motifs are mainly found in low intensity peptides. This over representation of consensus motifs for high intensity peptides is in agreement with the work of Hendriks *et al.*[20], and might be explained by UBC9 activity. Indeed, UBC9 is constitutively expressed in cells and is able to SUMOylate consensus sequences while non-canonical sequences requires E3 ligases for the proper transfer of SUMO onto their substrates[31].

We also identified 125 SUMOylated peptides containing phosphorylated residues (localization confidence >0.75) on serine (81 sites), threonine (9 sites; Fig. 2c, Supplementary Data 3). Phosphorylated Ser residues were primarily located 5 amino acids downstream of the SUMOylated residue, in agreement with previously published observations on phosphorylation dependent SUMOylation motif[8]. In contrast, no correlation with respect to the SUMOylated Lys was found for phosphorylated Thr residues.

**Concomitant profiling of SUMOylation and ubiquitylation.** One of the main advantages of this method over other large-scale approaches for SUMOylation site mapping is its compatibility with the concomitant identification of ubiquitylation sites. To show the feasibility of such an approach, we modified our protocol to enrich ubiquitylated and SUMOylated peptides on SUMOylated proteins (Fig. 3a). We opted to use this approach to profile SUMOylation and ubiquitylation changes following proteasome inhibition using MG132 for up to 8 h at 2 h intervals. Western blot analysis of the total cell extract revealed an overall increase of both SUMOylation and ubiquitylation levels

coupled with a ∼20% decrease in the pool of free SUMO or ubiquitin (Supplementary Fig. 7). However, NiNTA enrichment of SUMOylated proteins showed a significant enrichment of both SUMOylated and ubiquitylated proteins with a high degree of specificity in HEK293 cells expressing the SUMO3 mutant protein (Supplementary Fig. 7) whereas WT HEK293 cells did not show the corresponding enrichment (Supplementary Fig. 8a). Moreover, we confirmed that the ubiquitylation signal on our western blots did not arise only from ubiquitylation on polySUMO chains by performing an *in vitro* deSUMOylation assay (Supplementary Fig. 8b). Indeed, a portion of the ubiquitin signal on the western blot appears to stem from ubiquitin that is directly attached to the target substrate. We also assessed the extent of nonspecific binders arising from the NiNTA purification by comparing the number of identified SUMOylated protein in our comprehensive analysis to the raw NiNTA extract (Supplementary Fig. 8c). After NiNTA enrichment proteins were digested with trypsin and ubiquitylated peptides were enriched using the anti-K(GG) antibody cross linked to the agarose beads to limit nonspecific binding to NiNTA beads[32]. Since the occurrence of ubiquitylated peptides following NiNTA purification is expected to be much lower than SUMOylated peptides, we first performed immunoaffinity purification of the Gly-Gly containing tryptic peptides to minimize sample losses. Non-retained peptides found in the flow through were then immunopurified using our optimized SUMO enrichment protocol. Immunoprecipitated peptides eluted from both purifications were injected separately using LC-MS/MS. Supplementary Data 4 summarizes all SUMOylated and ubiquitylated peptides identified in this study. In total, we identified 1,616 SUMOylation sites on 918 proteins and 349 ubiquitylation sites on 194 proteins. Among the 349 ubiquitylated lysine residues, only 221 were identified as uniquely ubiquitylated, while 128 were modified by both ubiquitin and SUMO. Interestingly,

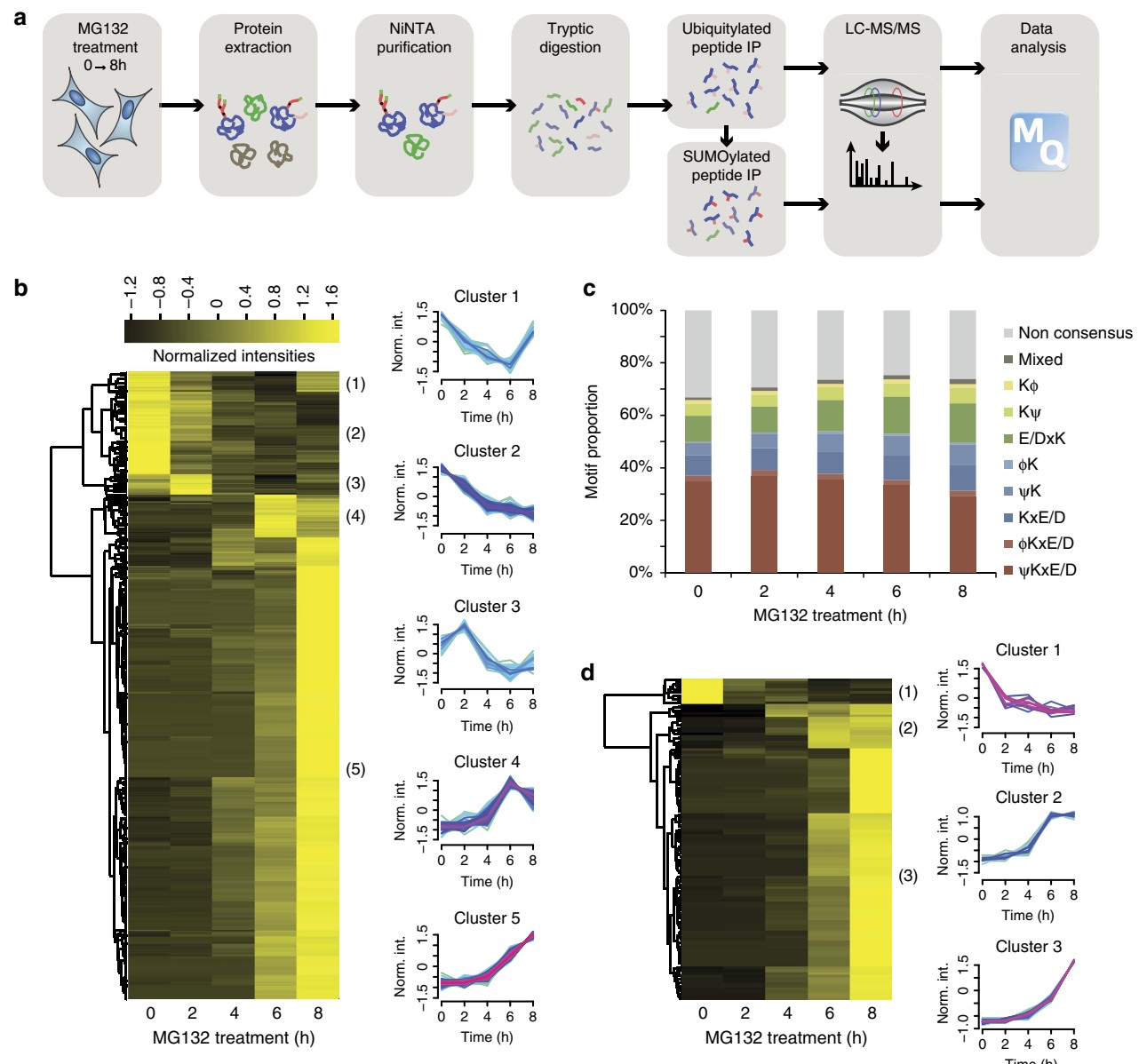

**Figure 3 | Temporal profiling of the SUMOylome and ubiquitylome in response to MG132 treatment.** (**a**) Overview of the strategy used to enrich SUMOylated and ubiquitylated peptides. Peptides arising from the digestion of SUMOylated proteins were subjected to a first peptide IP using the cross-linked anti-K-(GG) antibody. The flow through of the first IP was subjected to SUMOylated peptide enrichment with the anti-K-(NQTGG) antibody. The two resulting eluates were injected separately on the LC-MS/MS setup. (**b**) Heat map of normalized intensity for SUMOylated peptides as a function of MG132 treatment period. 1620 SUMO sites were identified with a localization confidence > 0.75. Peptide identifications were clustered using Fuzzy-C-means clustering. 676 sites were regulated and were divided in five groups based on their kinetic profiles. (**c**) Distribution of the proportion of SUMO motifs over time. A depletion of both non-consensus and consensus motifs and an increase of non-canonical motif over the 8 h time period was observed. (**d**) Heat map of normalized intensity for ubiquitylated peptides as a function of MG132 treatment period. 349 Ubi sites were identified with a localization confidence > 0.75. A total of 114 sites were regulated and were divided in 3 groups based on their kinetic profile.

113 of the 194 ubiquitylated proteins were not identified as SUMOylated. Since we performed an immunopurification against K-(GG) containing peptides after the NiNTA purification, we hypothesized that those unique proteins are SUMOylated on the polyubiquitin motif. Indeed, our data revealed more than 12 branched peptides on SUMO or ubiquitin (Supplementary Fig. 9), which also explains the concomitant SUMO/ubiquitin enrichment at the protein level. While the function of such chains remains mostly unknown, the ubiquitylation of polySUMOylation chains by the RING Finger Protein 4 (RNF4) was associated with the recruitment of ubiquitylated proteins for their subsequent proteasomal degradation[12]. Since we used MG132 for these experiments the RNF4-mediated degradation of these highly branched products was abrogated, leading to their accumulation in the cell.

Next, we quantified sites only observed in consecutive time points. This enabled the profiling of 778 SUMOylated peptides and 124 ubiquitylated peptides, corresponding to 676 and 114 sites, respectively (Fig. 3b,d). While western blot analysis reveals a global increase in the level of both SUMOylation and ubiquitylation (Supplementary Fig. 7), LC-MS/MS analysis revealed that a small proportion of SUMOylation

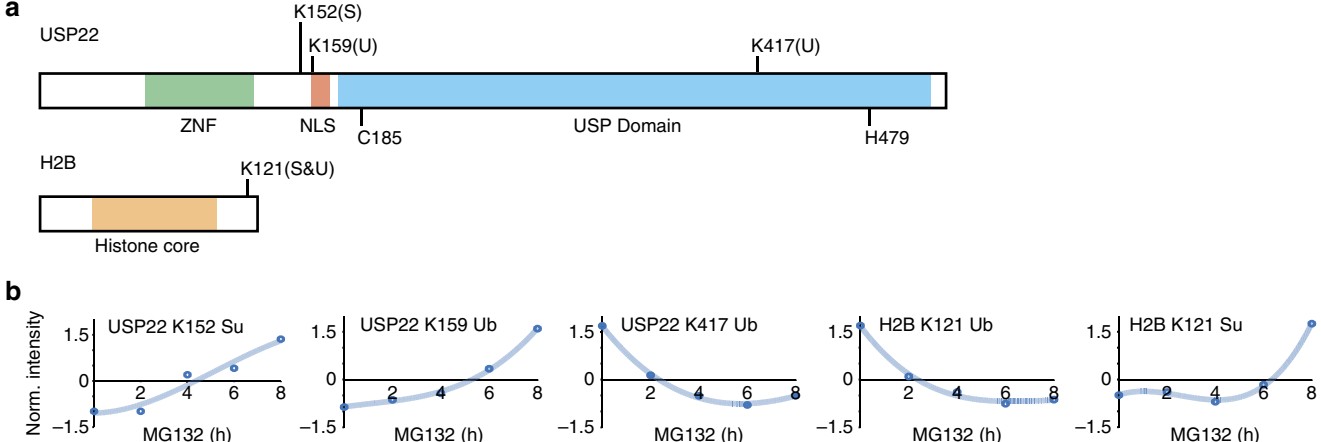

**Figure 4 | Crosstalk between SUMOylation and ubiquitylation levels on the histone deubiquitinase USP22 in response to MG132. (a)** Distribution of SUMOylation and ubiquitylation sites identified on the histone deubiquitinase USP22 and its substrate Histone H2B. The Zinc Finger domain of USP22 is shown in green, the potential nuclear localization sequence (NLS) in red, the catalytic domain in blue. The Histone core is depicted in orange. The lysine residues shown in bold depict the modified residues identified in this study. **(b)** Temporal profiles of the modified peptides of USP22 and H2B identified in the MG132 kinetic study.

(6.4%) and ubiquitylation (8.0%) profiles were down regulated on MG132 treatment. We also observed that the proportion of SUMOylation sites in consensus motifs increased for the first 2 h of the MG132 treatment and decreased then after at the expense of mixed and inverted motifs (Fig. 3c). We surmised that changes in the proportion of sequence motifs over time is partly explained by promiscuous SUMOylation, and/or the activation of SUMO E3 ligases brought on by extended exposure to MG132, though we cannot exclude a possible influence attributed to the use of SUMO3 mutation on the distribution of SUMO motifs for extended periods of MG132 treatment.

Among all identified proteins, some are directly linked to the regulation of UBLs. Indeed, we found that most of the SUMOylation machinery (e.g. SAE1, UBA2, UBC9, PIAS1) as well as some proteins from the ubiquitylation machinery (e.g UBE2O, RNF40, TRIM32, USP22), are modulated by ubiquitylation and/or SUMOylation. Moreover, 35% of all identified proteins contained at least two modification sites, but only 5 of these (UPS22, H2B, RDH11, HNRNPC and ZMYM4) showed reciprocal effects, where the increase of a modification at one site is accompanied with the simultaneous decrease at another site on the same protein (Supplementary Fig. 10). Interestingly, USP22 and its substrate H2B K121 (ref. 33) were among the proteins that showed a reciprocal effect (Fig. 4). On MG132 treatment, we observed an increase in the SUMOylation of USP22 at K152 and ubiquitylation at K157. These changes were accompanied by a decreased in the ubiquitylation of USP22 K417, a site located within its catalytic domain. In the case of H2B, we observed a decrease of ubiquitylation that was correlated with an increase of SUMOylation at K121. While the ubiquitylation status of H2B is critical for the transcriptional process[34], the function of H2B SUMOylation is presently unknown. Although USP22 is the only deubiquitinating enzyme for which we saw a reciprocal effect, we noted a similar behaviour for USP37 and its substrate c-Myc[35] (Supplementary Fig. 11). We observed an increase in the SUMOylation of USP37 at K452 located in its catalytic domain, a change that was correlated with an increase in the ubiquitylation of c-Myc at K148 and K389.

**SUMOylation and ubiquitylation of ribosome and proteasome.** To better understand the connectivity between SUMOylated and ubiquitylated proteins, we performed a string analysis (Fig. 5). As expected, the network of SUMOylated proteins shows a high degree of interconnectivity. Deeper exploration of this network revealed several protein complexes, including but not limited to the SWI-SNF complex (e.g. SMARCC1, SMARCC2 and SMARCD2), the ribosome (e.g. RPLs, RPSs, GNB2L) and the proteasome (e.g. PSMA, PSMB, PSM). Ribosome biogenesis was previously shown to be a SUMO-dependent mechanism[36]. We mapped all SUMOylation and ubiquitylation sites identified on the ribosome to its crystal structure (Supplementary Fig. 12a, PDB: 4UG0). In total, we identified 447 SUMOylation sites and 26 ubiquitylation sites on ribosomal proteins from kinetic and the 2DLC experiments. All the quantified SUMOylation and ubiquitylation sites on the ribosome subunits were upregulated after treatment with MG132. The structure highlights that the modification sites are located at the interface between subunits and possibly disrupt protein-protein and protein-RNA interactions. Since SUMOylation is believed to generally occur at a low stoichiometry (<5%), the effect of a single SUMOylation event on ribosomal activity may be negligible[37]. However, our kinetic studies have shown that 76 SUMO sites on 36 subunits are regulated by MG132. The compounded effect of several SUMOylation events occurring simultaneously may alter protein synthesis. This is consistent with the study reported by Kadlčíková et al.[38] which indicated that MG132 lead to a 10 to 20% decrease in protein synthesis.

The present study also highlighted that several proteasome subunits were extensively modified. Data from both 2DLC and kinetic experiments identified a total of 128 modified lysine residues on proteasome subunits with several sites showing a significant increase in their SUMOylation on MG132 treatment. We mapped all the modified lysine residues on the recently available proteasome crystal structure (Supplementary Fig. 12b, PDB: 4R3O). Among the 41 modified lysine residues, 36 are located at the surface of the 20S subunit while only 5 are observed inside the complex lumen (PSMA1 K115, PSMA2 K92, PSMA5 K91, PSMA6 K116 and PSMB3 K77). The distribution of modified residues on the outer surface of the proteasome subunits possibly implies that SUMOylation may affect protein-protein interactions. We hypothesized that the SUMOylation of the proteasome may act as an address tag to direct its subcellular localization. To confirm this proposal, we performed immunofluorescence analysis on 20S subunits in cells treated or not treated with MG132 (Supplementary Fig. 13).

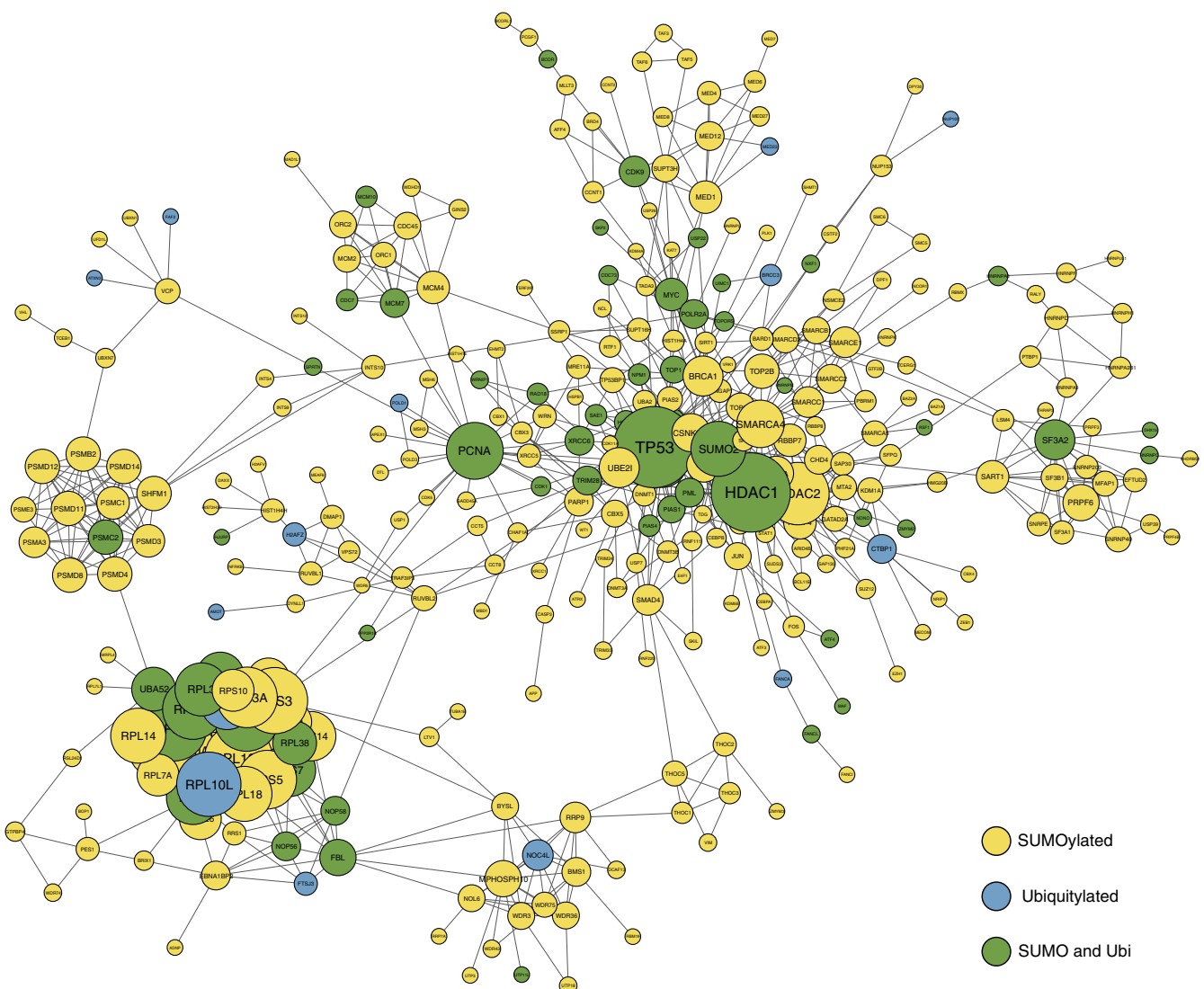

**Figure 5 | SUMOylated and ubiquitylated proteins show a high degree of interconnectivity.** Protein interaction network of all identified SUMOylation and ubiquitylation sites identifies in the kinetic experiment. Protein identified as only SUMOylated are represented in yellow, only ubiquitylated in bleu and identified as both in green. The size of each node represents the number of neighbours.

These experiments indicated that the endogenous 20S core is mainly cytoplasmic under untreated conditions, but is translocated to the nucleus on MG132 treatment where it partially co-localized with endogenous PML within the NBs (Supplementary Fig. 13a). Similar observations were obtained when these immunofluorescence experiments were performed with the β regulatory subunit of the 11S (Supplementary Fig. 13b). To investigate if the SUMO interacting motif (SIM) of PML is required for the recruitment of SUMOylated proteasome to PML NBs, we transfected HEK293-SUMO3m cells with PML IV or PML IV-SIM that is mutated in its SIM core sequence (VVVI hydrophobic amino acids; Fig. 6a). These experiments revealed that the 20S proteasome subunits co-localized with PML IV within the NBs in presence of MG132. In contrast, cells transfected with PML IV-SIM failed to recruit the proteasome under the same conditions (Fig. 6a). To confirm these results, we performed the same experiments in $PML-/-$ murine embryonic fibroblast (MEF) cells co-transfected with SUMO3 and PML IV or PML IV-SIM (Fig. 6c), which again showed that the SIM of PML was required for the recruitment of the 20S proteasome subunits to PML NBs.

These results also revealed that PML IV alone, in the absence of the others isoforms, was able to recruit the proteasome on MG132 treatment. The quantification of fluorescence intensities in both HEK293-SUMO3m and $PML-/-$ MEF cells revealed that the portion of 20S associated to PML NBs increased only in PML IV-expressing cells treated with MG132 (Fig. 6b,d). Indeed, Manders' co-localization coefficient were 0.58 and 0.65 for PML IV in HEK293 SUMO3m cells (Fig. 6b) and $PML-/-$ MEFs (Fig. 6d) treated with MG132, compared with 0.22 and 0.15 for PML IV-SIM in the same cells, respectively.

While we cannot entirely rule out the possibility that other modifications can affect the recruitment of the proteasome to PML NBs, the observation that SUMOylation mediates its recruitment is consistent with prior studies reporting the co-localisation of the proteasome and PML under different cellular stress including arsenic trioxide $(As_2O_3)$ (ref. 39). Altogether, these data suggest that the SUMOylation of the proteasome is a required signal for its recruitment via the SIM of PML to PML NBs where it facilitates protein degradation under different stress conditions.

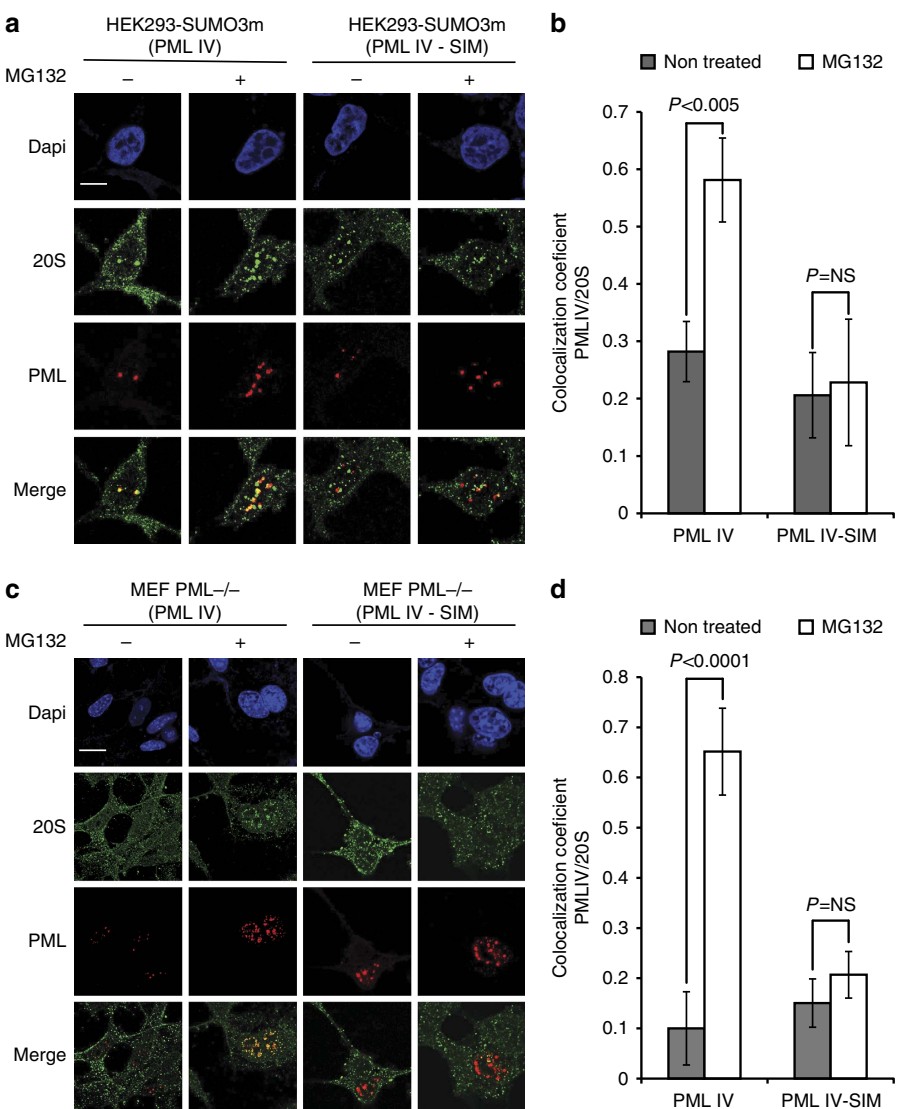

**Figure 6 | Immunofluorescence of 20S Proteasome and PML or PML-SIM under MG132 stress.** (**a**) HEK293-SUMO3m cells were transfected with PML IV or PML IV-SIM. Two days later they were treated with 10 μM MG132 for 6 h. Double immunofluorescence analyses were performed using monoclonal mouse anti-PML (red) and rabbit anti-20S (green) antibodies. (**b**) PML and 20S proteasome co-localization for PML and PML-SIM cells treated with MG132 from (a). The bar charts represent the ratio of the PML co-localized with the 20S proteasome, where error bars represent the standard deviation from 3 biological replicates for $n = 30$ cells. Experiments for (**c**) and (**d**) were performed as for (a) and (b) in PML $-/-$ MEFs co-transfected with SUMO3 and PML IV or PML IV-SIM, where error bars represent the s.d. from 5 biological replicates for $n = 60$ cells. Scale bars, 10 μm.

## Discussion

The optimized SUMOylated peptide immunopurification protocol provides an efficient approach to obtain unambiguous identification of SUMOylation sites on protein substrates. Altogether, this approach enabled the proteome-wide identification of 10,388 SUMO sites on 3,556 proteins in HEK293 cells. Approximately half of these sites were also identified in recent large-scale SUMO proteome analyses that used different cell lines and stimuli. Interestingly, SUMO consensus motifs were largely represented by highly abundant SUMOylated peptides while non-consensus motifs were mostly found for low abundance peptides, possibly highlighting promiscuous SUMOylation and/or the activation of SUMO E3 ligases taking place over extended periods of proteasome inhibition.

This method can be advantageously combined with immunoaffinity enrichment of ubiquitylated peptides to determine the extent of crosstalk between these modifications, and rationalize their functional relationships. For example, protein SUMOylation can mediate the recruitment of E3 ubiquitin ligases, which catalyse the ubiquitylation and proteasomal degradation of protein substrates[1]. A case in point is the ubiquitin ligase RNF4 that contains multiple SIMs and recognizes poly-SUMOylated proteins for their targeted proteasome degradation. RNF4 also associates with PML NBs under As$_2$O$_3$ treatment and proteasome inhibition[39–42]. Recently, additional polySUMO-binding proteins, including Arkadia, FLASH, C5orf25 and SOBP, have been identified through a computational string search[43]. As such, it is becoming clear that the PML/SUMO/poly-SUMO-dependent ubiquitin E3 ligases/proteasome complex constitutes a general mechanism to destroy NB-targeted proteins. While PML NBs are known to be a site of protein degradation, the exact mechanism by which proteasomes are recruited to these nuclear structures is not entirely known. Using a quantitative proteomics approach,

we uncovered that several proteasome subunits are SUMOylated on MG132 treatment, and that this modification is required for the recruitment of the proteasome to PML NBs *via* the SIM of PML. Accordingly, PML lacking the SIM sequence is resistant to As$_2$O$_3$-induced PML degradation due to its inability to recruit the proteasome components in PML NBs[39]. It is noteworthy that previous reports highlighted the co-localization of the proteasome with lytic vesicles such as autophagosomes[44], phagosomes[45] and exosomes[46] when cells are exposed to different stimuli. Proteasome SUMOylation could thus provide a convenient recognition signal to recruit the degradation machinery at primary sites of ubiquitin-mediated proteolysis.

The dynamic profiling of protein SUMOylation and ubiquitylation also revealed unexpected crosstalk between UBL modifiers. Moreover, we observed that USP22 and USP37 can be modified by SUMO and ubiquitin at various sites simultaneously and are regulated by MG132, though their functions remain to be identified. Deubiquitylation of USP22 at K417, a site located within its catalytic domain, led to the decreased ubiquitylation of H2B at K121 and its subsequent SUMOylation. Similarly, the increased SUMOylation of USP37 at its catalytic site K452 correlated with an increase in ubiquitylation of its substrate c-Myc at K148 and K389. These observations suggest that the SUMOylation or ubiquitylation of lysine residues within the catalytic domain of deubiquitinating enzymes could represent a novel inhibitory mechanism mediating their enzymatic activities.

These results demonstrate that immunoaffinity methods enabling the enrichment of SUMOylated and ubiquitylated peptides can be used to isolate UBL-modified substrate lysine residues present at low abundance in human cells. This approach opens up new avenues for the identification of protein substrates, their site-specific modifications, the interplay between SUMOylation and ubiquitylation, and their regulation on different environmental conditions.

## Methods

**Materials.** SCX spin tips (SP-155) were purchased from Protea (Morgantown, WV). PureProteome Protein A/G Mix Magnetic Beads (LSKMAGAG10) was purchased from EMD Millipore (Ottawa, ON, Canada). Dimethyl pimelimidate (DMP) cross-linking reagent (21,666) was obtained from ThermoFisher scientific (Burlington, ON, Canada). The custom anti-K(NQTGG) SUMO remnant antibody was obtained from Epitomics/Abcam (Burlingame, CA). PTMScan Ubiquitin Remnant Motif (K-ε-GG) Kit (5,562) was purchased from Cell Signalling Technologies (Danvers, MA). Modified porcine sequencing grade modified Trypsin was obtained from Promega (Madison, WI, USA). Acetonitrile was purchased from Fisher Scientific (Whitby, ON, Canada). Ammonium bicarbonate and formic acid were obtained from EM Science (Mississauga, ON, Canada). 2-mercaptoethanol, ammonium hydroxide, trifluoroacetic acid, 2- chloroacetamide, protease inhibitor cocktail (4-(2-aminoethyl)benzenesulfonyl fluoride, pepstatinA, E-64, bestatin, leupeptin and aprotinin), phosphatase inhibitor cocktail (sodium vanadate, sodium molybdate, sodium tartrate and imidazole) were purchased from Sigma-Aldrich (Oakville, ON, Canada). Bradford protein reagent was obtained from Bio-Rad (Mississauga, ON, Canada). Tris base was purchased from EMD Omnipur (Lawrence, KS). PBS was obtained from HyClone (Thermo Scientific, Logan, UT). Oasis hydrophilic-lipophilic balance (HLB) cartridges (1cc, 30 mg) were purchased from Waters (Milford, MA). ECL chemiluminescence detection system was purchased from Amersham Pharmacia Biotech (Montréal, QC, Canada). Mouse anti-PML (sc-966) was obtained from Santa Cruz. Rabbit anti-proteasome activator 11Sβ (BML-PW8240) and anti-20 S (BML-PW8155) antibodies were purchased from Enzo life sciences (Brockville, ON, Canada). Goat anti-mouse IgG and goat anti-rabbit IgG secondary antibodies were purchased from Millipore (Ottawa, ON, Canada). Solvents for chromatographic analysis were all high-performance liquid chromatography (HPLC) grade (Fisher Scientific and in-house Milli-Q water). Capillary HPLC columns for nano-LC-MS were packed in-house using Jupiter C18 (3 μm) particles from Phenomenex (Torrance, CA), and fused silica tubing from Polymicro Technologies (Phoenix, AZ). PML IV (accession number NP_002666.1) was used in this study. PML IV-SIM mutant was obtained by mutating the four hydrophobic residues from the SIM (VVVI) to (AAAS). Cell viability was evaluated using the Cell Proliferation Kit I (MTT) purchased from Roche (Mississauga, ON, Canada).

**Cell culture.** HEK293 stably expressing 6xHisSUMO-3-Q87R-Q88N cells (SUMO3m) were obtained by transfecting HEK293 cells with our SUMOm gene in pcDNA3 and subsequent neomycin selection (0.5 mg ml$^{-1}$)[16]. MEFs PML −/− were derived from PML knock out mice[47]. Cells were grown in DMEM supplemented with 10% FBS, 1% L-Glu, 1% penicillin/streptomycin and 750 μg ml$^{-1}$ geneticin at 37 °C in a 5% CO$_2$ atmosphere. At 70% confluence, cells were either mock treated or treated with 10 μM MG132 for up to 16 h. For the kinetic profiling, a single T175 flask has been used per time point.

**Protein purification and digestion.** HEK293 SUMO3m cells were washed twice with ice cold PBS and lysed in NiNTA denaturing incubation buffer (6 M Guanidinium HCl, 100 mM NaH2PO4, 10 mM Tris-HCl, 20 mM 2-Chloroacetamide, 5 mM 2-Mercaptoethanol, pH = 8) and sonicated. Total protein content was determined using micro Bradford assay. For each condition, 4 mg of total cell extract (TCE) were incubated with 80 μL of NiNTA beads for 16 h at 4 °C. NiNTA beads were washed once with 1 mL of NiNTA denaturing incubation buffer, 5 times with 1 mL of NiNTA denaturing washing buffer (8 M urea, 100 mM NaH$_2$PO$_4$, 10 mM Tris-HCl, 20 mM imidazole, 5 mM 2-Mercaptoethanol, 20 mM Chloroacetamide, pH = 6.3) and finally twice with 1 mL of 100 mM ammonium bicarbonate. The protein content was determined by micro Bradford assay. On beads protein digestion was performed by adding trypsin to a ratio 1:50 by weight (Trypsin:Protein extract) for 4 h at 37 °C. Resulting NiNTA enriched digests were acidified by adding trifluoroacetic acid to a final concentration of 1%, desalted on HLB cartridges as per manufacturer's instructions and eluted in LoBind tubes before being dried down by Speed Vac.

**Antibody cross-linking.** PureProteome protein A/G magnetic beads were equilibrated with anti-K(NQTGG) antibody (2 μg of antibody per μl of slurry) for 1 h at 4 °C in PBS. Saturated beads were washed 3 times with 200 mM triethanolamide pH 8.3. For crosslinking, 10 μl of 5 mM DMP in 200 mM triethanolamide pH 8.3 was added per μl of slurry and incubated 1 h at room temperature. The reaction was quenched for 30 minutes by adding 5% (v/v) of 1 M Tris-HCl pH = 8. Cross-linked beads were washed 3 times with ice cold PBS and once with PBS containing 50% Glycerol and stored at −20 °C for future use.

**Dual peptide IP.** For the ubiquitin IP, dried digests were reconstituted in 500 μL of PBS and supplemented with 31 μg of anti-K(GG) antibody. The flow through was kept for the anti-K(NQTGG) IP. Anti-K(GG) antibody bound beads were washed three times with PBS. Ubiquitylated peptides were eluted 3 times with 200 μL of 0.2% formic acid in water and filtered through a 0.45 μm spin tube. Eluted peptides were dried down by speed vac and stored at −80 °C for MS analysis. For the anti-K-(NQTGG) IP, ubiquitin IP flow throughs were diluted with 90% Glycerol in PBS to obtain a final glycerol concentration of 50%, supplemented with 20 μg of cross-linked anti-K-(NQTGG) (1:2 by weight (Antibody:NiNTA digest)) and incubated for 1 h at 4 °C. anti-K-(NQTGG) antibody bound beads were washed three times with 1 ml of 1 × PBS, twice with 1 ml of 0.1X PBS and once with water. SUMO peptides were eluted 4 times with 100 μl of 0.2% formic acid in water and filtered through a 0.45 μm spin tube. Eluted peptides were dried down by speed vac and stored at −80 °C for MS analysis.

**SCX fractionation.** Peptides were reconstituted in water containing 15% acetonitrile and 0.2% formic acid and loaded on conditioned Protea SCX tips. Peptides were eluted with ammonium formate pulses at 50, 75, 100, 300, 600 and 1,500 mM in 15% acetonitrile, pH = 3 (pH adjusted with formic acid). Salts factions were dried down by speed vac.

**Mass spectrometry analysis.** Peptides were reconstituted in water containing 0.2% formic acid and analysed by nanoflow-LC-MS/MS using an Orbitrap Fusion Mass spectrometer (Thermo Scientific) coupled to a Proxeon Easy-nLC 1000. Samples were injected on a 300 μm ID × 5 mm trap and separated on a 150 μm × 20 cm nano LC column (Jupiter C18, 3 μm, 300 Å, Phenomenex). The separation was performed on a linear gradient from 7 to 30% acetonitrile, 0.2% formic acid over 105 minutes at 600 nl per min. Full MS scans were acquired from m/z 350 to m/z 1,500 at resolution 120,000 at m/z 200, with a target AGC of 1E6 and a maximum injection time of 200 ms. MS/MS scans were acquired in HCD mode with a normalized collision energy of 25 and resolution 30,000 using a Top 3 s method, with a target AGC of 5E3 and a maximum injection time of 3,000 ms. The MS/MS triggering threshold was set at 1E5 and the dynamic exclusion of previously acquired precursor was enabled for 20 s within a mass range of ± 0.8 Da.

**Data processing.** Peptide identification was performed using MaxQuant (version 1.5.1.2)[48]. MS/MS spectra were searched against Uniprot/SwissProt database including Isoforms (released on 10 March 2015). The maximum missed cleavage sites for trypsin was set to 2. Carbamydomethylation (C) was

set as fixed modification and acetylation (Protein N term), phosphorylation (STY), oxidation (M), deamination (NQ), GG (K) and NQTGG (K) were set as variable modifications. The false discovery rate for peptide, protein, and site identification was 1%. SUMO sites with a localization probability of $> 0.75$ were retained. Peptide quantification was achieved with match between runs enabled. To regroup identified site according to their kinetic profiles, Fuzzy-C-means algorithm[49] platform developed in our lab[50] was used. Briefly, data processing was performed in an R environment (www.r-project.org) with the MFuzz package[51]. Sites that were not detected in at least 3 time point or that were detected in non-contiguous time points were automatically rejected. Sites with a membership value higher than 0.5 were retained for downstream analysis. Pearson coefficients for biological reproducibility were obtained by plotting the fold changes of MG132/control for the three biological replicates.

**Bioinformatics analysis.** Sequence windows spanning ± 6 amino acids around the modified lysine residue obtained in Andromeda were submitted to Motif X (ref. 30). For peptide sequences corresponding to multiple proteins, only the leading sequence was submitted. Network analysis was done using String database[52] with highest confidence (score > 0.9) and experimental data only.

**Fluorescence imaging and co-localization analysis.** Cells grown on glass coverslip were fixed with cold acetone for 10 min at − 20 °C. Cells were prepared for immunofluorescence staining and analysed by confocal microscopy. PML was detected with mouse anti-PML (sc-966) antibody and the corresponding anti-IgG antibody conjugated to Alexa 594. The rabbit polyclonal antibodies against proteasome 20 S core and against regulatory subunit of 11 S β were used for detection of proteasome components followed by Alexa 488. The cells were washed in PBS, stained with 4,6-diamidino-2-phenylindole and mounted in Fluoromount-G medium. Images were digitally acquired with a Zeiss LSM 710 confocal Microscope. For quantification of 20 S co-localization with PML IV or PML IV-SIM on MG132 treatment, the JACoP plugin in ImageJ software[53] was used and the Manders' coefficient was calculated, $n = 60$ cells per condition.

**Western blot.** An aliquot of 10 μg TCE prepared in NiNTA denaturing incubation buffer were diluted in Laemmli buffer (10% (w/v) glycerol, 2% SDS, 10% (v/v) 2-mercaptoethanol and 0.0625 M Tris-HCl, pH = 6.8), boiled for 10 min and separated on a 4–12% SDS-PAGE followed by transfer onto nitrocellulose membranes. For NiNTA purified material, 100 μg of TCE from WT cells or 10 μg of TCE from SUMO3m cells were purified before on NiNTA beads, eluted in Laemmli buffer and boiled for 10 minutes before SDS-PAGE. Before blocking the membrane for 1 h with 5% non-fat milk in TBST (tris-buffered saline with Tween 20), membranes were briefly stained with 0.1% Ponceau-S in 5% acetic acid to represent total protein content. Membranes were subsequently probed with the primary antibody, as indicated, in blocking solution at 4 °C for 16 h. (SUMO2/3, 1:2,000, Zymed; Ubiquitin, 1:200, Santa Cruz; Histone H3, 1:1,000, Cell Signalling Technologies) The membranes were incubated with secondary antibodies (goat anti-rabbit HRP, EMD Millipore, 1:5,000 and goat anti-mouse HRP, EMD Millipore, 1:5,000) for 1 h at room temperature. Membranes were washed three times with TBST for 10 minutes each. Membranes were revealed using ECL (GE healthcare) as per the manufacturer's instructions, and chemiluminescence was captured on Blue Ray film.

**Data availability.** The raw data that support the findings of this study are available from Peptide Atlas, http://www.peptideatlas.org with the accession code PASS00896. The additional data that support the findings of this study are available from the corresponding author on request.

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

## Acknowledgements

This work was funded in part by the Natural Sciences and Engineering Research Council (NSERC; P.T.), the Agence Nationale de la Recherche (ANR11BSV30028) and the Agence Nationale de Recherches sur le SIDA et les hépatites virales (ANRS; M.K.C.-A). F.L. was supported by a scholarship from the faculty of graduate studies of the Université de Montréal and F.P.M. is the recipient of postdoctoral fellowship from NSERC. The Institute for Research in Immunology and Cancer (IRIC) receives infrastructure support from Genome Canada, the Canadian Center of Excellence in Commercialization and Research, the Canadian Foundation for Innovation, and the Fonds de recherche du Québec-Santé (FRQS). We would like to thank Peter Kubiniok for assistance with bioinformatics analyses.

## Author contributions

F.L., F.P.M., G.M. performed experiments and analysed data; F.L., F.P.M., M.K.C.-A. and P.T. wrote the manuscript; P.T. developed the concept and managed the project.

## Additional information

**Competing financial interests**: The authors declare no competing financial interests.

**How to cite this article**: Lamoliatte, F. *et al.* Uncovering the SUMOylation and ubiquitylation crosstalk in human cells using sequential peptide immunopurification. *Nat. Commun.* **8,** 14109 doi: 10.1038/ncomms14109 (2017).

**Publisher's note**: 

