## [Peer Review File · Nature Communications]

Reviewers' Comments:

Reviewer #1 (Remarks to the Author):

Lamoliatte et al describe a workflow that allows the large scale site mapping of SUMO modification sites and the identification of sites that are thought to be both SUMO and ubiquitin modified. The significant finding of this paper is the ability to identify in the order of 10000 SUMO sites in a single experiment from relatively modest starting material amounts. However it was not clear clear what optimizations or novel methodologies contributed most to this improvement over the previous method from the same group which identified 1000 sites. It seems from the text it is the SCX fractionation of peptides prior to MS analysis that did it, but this is not investigated experimentally, while other modifications are altered, apparently with little consequence.

The other major claim of the paper is the ability to analyse SUMO and Ub sites from the same preps. But much of the 'new biology' is in relation only to SUMO sites (ribosomes/proteasome). This is already available. What little analysis there is regarding co-regulation by SUMO and ubiquitin is quite weak, based largely upon observed changes rather than direct relationships. These experiments revealed that the 20S proteasome subunits co-localized with PML IV within the NBs in presence of MG132. In contrast, cells transfected with PML IV-SIM failed to recruit the proteasome under the same conditions (Fig. 5d). The images in this figure are entirely unconvincing and do not support the conclusion that SUMO modified proteasomes are recruited to PML bodies. It was not clear that the PML IV was transfected into PML^{-/-} cells. If it was not transfected into PML^{-/-} cells why did the wild type PML not recruit the SUMO modified proteasome ? In these types of experiments it is critical to quantitatively assess the amount of colocalisation in each cell and count at least 100 cells.

They also observed that USP22 and USP37 were modified by SUMO and ubiquitin, and suggested that these modifications could regulate their deubiquitinase activities. There is no direct evidence for this statement, simply correlations. What would be required here is an in vitro experiment that measures the deubiquitinating activity of the protease when it is SUMO modified on the indicated site.

Specific points

1. To increase SUMO proteome coverage, the authors fractionated the samples using strong cation exchange (SCX) spin tips prior to LC-MS/MS analysis. This seems to have been the

biggest single contributor to the ~10x increase in site IDs over their previous study (and previous experiments here). Isn't this the factor that should be 'tested' for its ability to increase ID rates.

Meaning, why don't they do an experiment comparing unfractionated with SCX fractionated samples? Also it would be good for the authors to explain why they chose to fractionate peptides further considering peptide IPs tend to be already quite low in peptide amounts? Can the authors provide data like TIC traces to explain their motivations and show results of SCX fractionation?

2. The authors claim to have developed a high sensitivity MS method on the Orbitrap Fusion, If this is the case they should, using exactly the same sample, compare this new method with their previous to assess how much better it is.

3. The authors evaluated the reproducibility using "three biological replicates with three technical replicates each". However they need to clarify exactly what they mean by technical and biological replicates. Are technical replicates the same peptide prep injected into the mass spec three times or is it the same set of cells divided into 3 and carried through the workflow separately.

4. Precisely how the authors shortlisted sites regarded as 'true' SUMOylation sites is unclear. What were the MaxQuant settings for FDR filtering? Did they just take the MaxQuant modified peptides output or were further levels of filtering applied? It is important to ascertain how confident the authors were in their site identifications. This is particularly important as many people in the SUMO field are not convinced SUMO4 is actually expressed. My assessment of the data on SUMO4 is that it is a pseudogene. However taking the data here at face value would suggest that SUMO4 is indeed expressed. Thus it is important that the identifications of modifications to SUMO4 reported here are put beyond reasonable doubt. Thus annotated spectra of the SUMOylation and ubiquitination sites of SUMO4 should be provided. This is especially important given that SUMO4 differs from other SUMOs by only a few amino-acids and many of the reported SUMO-4 peptides have the same Andromeda score as peptides from SUMO-2 and/or SUMO-3 implying they share the same identifying spectra, and so may be false identifications. This is clear from the supplied Excel data files where SUMO4 ids are often grouped into the same protein groups as SUMO2/SUMO3 IDs at conserved lysines.

5. The authors purified SUMO modified proteins on the basis of the 6His tag on the expressed mutant version of SUMO and used this material to enrich for SUMO sites and ubiquitin sites . However more than half of the ubiquitinated proteins identified were not shown to be SUMO modified. The authors hypothesize that those unique proteins are SUMOylated on the polyubiquitin motif. This may be the case but it is also possible that this will arise from non-specific NiNTA binding proteins which make up the majority of NiNTA elutions. This would be an interesting point to sort out and it could be done by simply treating the eluted material with a recombinant SENP and analyzing the ubiquitinated proteins by Western blotting.

6. It is suggested that "SUMOylation of ribosomal subunits may prevent the formation of the protein complex, thus altering protein synthesis". If SUMO is to block something then the biological significance depends on stoichiometry. If only a small proportion of the protein is

SUMO modified, then only a small proportion will be blocked from complex assembly. The authors need to consider/counter this.

Reviewer #2 (Remarks to the Author):

This paper outlines an affinity-purification and mass spectrometry (MS)-based approach for identifying proteins modified by both SUMO and ubiquitin (Ub). Using this system, the authors seek to clarify important outstanding questions about the nature of cross-talk between these two post-translational modifications which both occur on lysine residues. Investigating SUMO-Ub crosstalk has been challenging due to the C-terminal sequence of SUMO yielding a remnant that in its endogenous form is incompatible with large scale MS analysis. The paper appropriately cites the existing literature and the several available methods for enriching SUMO-modified proteins. Taking advantage of a mutant SUMO system and SUMO remnant antibody previously developed by these authors, Lamoliatte and colleagues perform a series of optimization studies to fine tune the methods for capturing SUMO3 remnant peptides by 2-stage immunoaffinity enrichment. In the first stage, Sumoylated proteins are captured by IMAC using the His-Tag encoded to the mutant SUMO3, followed by immunoaffinity enrichment of remnant peptides from either Ub or exogenous SUMO3. The first several figures and much of the supplement thoroughly details these methods development studies that are well done and will be interesting to proteomics specialists directly involved in this field. HEK293 cells are an ideal model for the methods development given the number of MS proteomics papers identifying both sumoylation and ubiquitylation sites in this cell line. For biological follow up, the current study investigates SUMOylation of deubiquitinase enzymes and the proteasome itself as potential nodes for crosstalk between the two signaling pathways. The strength of this paper is also its weakness when it comes to the current journal. The manuscript currently reads with a focus more suited to a proteomics-focused journal (i.e. not initiating the biology story until three pages into the Results section (bottom p7)). The premise that proteasome inhibition can profoundly alter crosstalk between Ub and SUMO3 is interesting, and has not been studied in this level of detail. That said, key elements of the story remain to be developed and would be needed to justify publication in Nature Communications. Specific issues and minor concerns are listed below.

- As general feedback, many of the figure labels are too small and/or illegible. This is the case for axis labels throughout Figs 2-5. It is particularly difficult to see the image labels in Fig 5D-E. Please improve the readability of figure legends throughout the manuscript and supplement.
- Proteasome inhibitors like MG132 influence many aspects of cellular signaling. It wasn't clear

from the background and results text why the authors chose to perform the comprehensive proteomics analysis of Sumo-Ub crosstalk on MG132 treated cells? Was this to enhance sensitivity for the analytical method by increasing SUMO-modifications in cells? Or in contrast, was it to reveal the nature of the underlying biology between SUMO, Ub and the proteasome. It seemed like it was both, but the experimental design and certain data interpretation are predicated on this rationale. Some additional context would be helpful in framing this study.

- Given that MG132 treatment depletes free ubiquitin in the cell, it is possible that SUMO is being conjugated to substrates and incorporated into ubiquitin-chains to serve as an alternate degradation signal. Work from Kim et.al. Mol Cell showed precedent for this with Nedd8 under conditions of proteasome inhibition. As such, the nature of SUMO-Ub crosstalk is highly dependent on the state of the Ub pool. That said, this topic was not discussed or considered as part of the experimental design or discussed when describing the results. Moreover, the blots in Fig. S7 do not depict 8Kd free Ub, which is critical for interpreting the decrease in signal at the 8 hr timepoint in the middle panel. Worth noting that historically, special methods are sometimes required to obtain good signal for mono-Ub, as most anti-Ub reagents detect polyUb better than the mono-Ub species.

- On a related note...are the cells still alive at the 8hr timepoint after 10 uM MG132? This is a significant exposure that in past experience may be expected to cause cells to round up and come off the cell culture dishes. What is the nature of the His-SUMO3 expressing cells at the various timepoints characterized. Does expression of His-SUMO3 construct alter cell sensitivity to the time or concentration of proteasome inhibitors required to cause cell death?

- The term "crosslinking" is used to describe the method of covalently binding the antibody to the resin and preventing its elution at the end of the immunoaffinity enrichment step. Yet within the text, the authors also use this phrase to mean something different, referring to heterogeneous chains comprised of mixtures of ubiquitin and SUMO. The text should more clearly differentiate between these two and not use a single phrase interchangeably.

- The authors surmise that changes in the SUMO sequence motif preference is explained by promiscuous sumoylation by Ubc9 and/or altered SUMO-E3 ligase activity following through MG132 treatment. Are the authors concerned that the motif preference could derive from a bias introduced by the mutant form of SUMO? Could it be that the introduced mutations, which occur in close proximity to the substrate motif, could be altering the conclusions drawn from these data? At the least, some discussion is required on this point.

- Are the authors concerned that the antibody used to IP SUMO-remnant peptides could preferentially bind peptides with certain sub-sequences? Has the binding specificity of this 'motif-specific' antibody been thoroughly tested?

- A more thorough accounting of the statistical tests should be provided in the methods section or supplement. What do the author mean when stating "reproducible label-free quantification was obtained for all biological replicates with a Pearson coefficient higher than 0.9". The numbers/method underlying this statement should be more clearly presented. Similarly, saying that "data revealed a sizeable amount cross-linked peptide" feels like an attempt to make a quantitative statement. A well defined comparison would be more appropriate.

- Figure 5e is not referenced in the figure text...looks like a is numbering/ duplication of Fig 5D on p11.

- Fig 5 is critical biology that is presented as the main novel element of the paper. Indeed, recruitment of proteasomes to the nuclear body via PML under conditions of stress seems potentially meaningful. For that reason, it is not sufficient to provide single cell, representative images to support these foundational claims. Systematically counted image data with sufficient n to support statistical testing would typically be required to support a claim of this sort.

- On p11 when discussing proteasome subunits, the text mistakenly notes "SUMOylation of the ribosome".

- Along with monoUb discussed above, histone 2A-Ub is an abundant storage depot for Ub that is also depleted upon prolonged proteasome inhibition. It appears that the blots in S7 are showing this, but it should be made more clear by capturing an extended exposure for the smaller molecular weight region of the gel. This becomes more important given that USP22, a protein highlighted as a critical crosstalk node between SUMO and Ub, is part of the SAGA complex that removed Ub from H2A. If the authors could expand upon this connection between USP22 SUMOylation, deubiquitination of H2A, the ubiquitin pools in relation to proteasome inhibitor sensitivity, this would be a meaningful biology result.

- Figure 5e is not referenced in the figure text

- 'Lysines' (or any amino acid) should be referred to as 'lysine residues'

- Some phrases in the text are unclear and the manuscript would be easier to read and understand if these sentences were edited: for example;

o Page 4 "several unsuspected crosstalks"

o Page 6 "we also determined the number of identification obtained with these two strategies,"

Reviewer #3 (Remarks to the Author):

Lamoliatte et al present "Uncovering Sumoylation and Ubiquitylation Crosstalks in Human Cells Using Sequential Peptide Immunopurification." The authors present methods for studying Ubiquitylation on Sumoylated proteins using a sequential enrichment strategy where they first enrich sumoylated proteins and then enrich ubiquitylated peptides and sumoylated peptides. The paper focuses on technical improvements that allow for the identification of ~10,000 sumo sites from cells expressing a 6xHis Sumo mutant. The results shown in this paper would be much more impactful if endogenous sumo sites were mapped instead of sites arising from the mutant 6xHis-Sumo and if cells had not been pre-treated with MG-132 as this causes significant cellular stress and increases sumo and ub stoichiometry. This study would be better suited for a proteomic technical journal as the most impactful parts of the paper are focused on the details of the enrichment.

Some specific points:

- The authors state that they enrich ubiquitylated and sumoylated peptides from sumoylated proteins. This statement is concerning. How can the authors be certain that the ubiquitin sites identified come from sumoylated proteins and not background, non-sumoylated proteins that came down in the NI-NTA enrichment.
- What is the specificity of the NI-NTA protein level enrichment? How would one measure that?
- Why is sumo protein level enrichment needed? Is the anti-sumo remnant antibody not specific enough without an initial enrichment step? If so, this makes the workflow more tedious and prone to sample loss.
- How many peptides harbored both KGG and sumo sites?
- Why is KGG enrichment completed first? There is no explanation for why this workflow was chosen.
- Why is the yield of sites from the sequential sumo and ubiquitin experiments low (1620 sumo sites single time point and 778 in consecutive time points) compared to the initial experiments described in the paper (~10K sites)? Was the optimized workflow not used?
- If cells were not pre-treated with MG-132 to increase sumo stoichiometry, would this workflow still enable enrichment of >1000 or >5000 sumo sites? What protein input amount would be needed?

Response to review of Manuscript "Uncovering the SUMOylation and Ubiquitylation Crosstalks in Human Cells Using Sequential Peptide Immunopurification" - **NCOMMS-16-14810**

Italic highlights response on specific comments raised by the reviewers

Bold highlights places where modifications have been made in the revised manuscript, also highlighted in yellow in the revised manuscript

We thank all reviewers for their valuable comments and constructive criticisms to improve the quality of our manuscript.

Reviewer #1:

1. The significant finding of this paper is the ability to identify in the order of 10000 SUMO sites in a single experiment from relatively modest starting material amounts. However it was not clear what optimizations or novel methodologies contributed most to this improvement over the previous method from the same group which identified 1000 sites. It seems from the text it is the SCX fractionation of peptides prior to MS analysis that did it, but this is not investigated experimentally, while other modifications are altered, apparently with little consequence.

To increase SUMO proteome coverage, the authors fractionated the samples using strong cation exchange (SCX) spin tips prior to LC-MS/MS analysis. This seems to have been the biggest single contributor to the ~10x increase in site IDs over their previous study (and previous experiments here). Isn't this the factor that should be 'tested' for it's ability to increase ID rates. Meaning, why don't they do an experiment comparing unfractionated with SCX fractionated samples? Also it would be good for the authors to explain why they chose to fractionate peptides further considering peptide IPs tend to be already quite low in peptide amounts? Can the authors provide data like TIC traces to explain their motivations and show results of SCX fractionation?

The reviewer highlights an important issue that requires further clarification. We concur that SCX fractionation reduces sample complexity and increase the depth and comprehensiveness of SUMO proteome analyses. As indicated on p.7 of the revised manuscript, we used SCX to minimize sample overloading on reverse phase pre-column. The limited sample capacity of the reverse phase column is highlighted in Fig. S3, where the proportion of non-SUMOylated peptides is progressively increasing with sample loading. The displacement of SUMOylated peptides by non-SUMOylated peptides with sample loading is further supported by their corresponding hydrophobic and hydrophilic scores as illustrated in Fig. S4.

It is noteworthy that we previously used SCX fractionation in our 2014 paper and obtained only ~1000 sites with 12% SUMO enrichment, thus supporting that data obtained here attest of the improved enrichment protocol of our new method. Indeed, our optimized protocol enabled the identification of ~10000 SUMO sites with enrichment levels higher than 70 %. Improvement in enrichment levels were achieved through optimization of several steps including MS/MS acquisition, antibody coupling to beads ligand:antibody ratio, and binding/elution buffers as described in p.5-7.

To clarify the advantages of the present approach we added the following text to the revised manuscript:

On p. 6: We noted that the proportion of non-SUMOylated peptides increased significantly beyond 4 mg of protein digest, possibly through the displacement of hydrophilic SUMOylated peptides from the reverse phase precolumn by hydrophobic tryptic peptides. Closer examination of the immunopurified extract revealed that SUMOylated peptides are more hydrophilic with a lower proportion of aromatic and aliphatic residues compared to their non-modified counterparts (Supplementary Fig. 4).

On p.7: To increase SUMO proteome coverage and minimize sample overloading on the reverse phase pre-column for injections exceeding 4 mg of digest from Ni-NTA purified proteins, we fractionated the samples using strong cation exchange (SCX) spin tips prior to LC-MS/MS analysis.

On p.7: This approach yielded an unprecedented number of identification with 9816 SUMO sites on 3405 proteins corresponding to enrichment levels above 70% (Supplementary Table 1 and 2). In contrast, the same analysis performed using reverse phase LC-MS/MS only yielded 1170 SUMO sites (Supplementary Fig. 3c). It is noteworthy, that 2D-LC-MS/MS experiments performed using the in solution antibody binding protocol previously identified only 954 SUMO sites¹⁷.

The reviewer would like the LC chromatograms of the SCX fractionated samples be included with the article. We uploaded all the raw files directly to Peptide Atlas as we did for our previous article so that others may benefit from the availability of the raw LC-MS/MS datasets. Reviewers can access the data at <ftp://PASS00896:Thibault01@ftp.peptideatlas.org/>. We added the following text on p. 25 of the revised manuscript:

The proteomic raw data associated with this manuscript has been uploaded to Peptide Atlas (<http://www.peptideatlas.org>) under accession code PASS00896.

2. These experiments revealed that the 20S proteasome subunits co-localized with PML IV within the NBs in presence of MG132. In contrast, cells transfected with PML IV-SIM failed to recruit the proteasome under the same conditions (Fig. 5d). The images in this figure are entirely unconvincing and do not support the conclusion that SUMO modified proteasomes are recruited to PML bodies. It was not clear that the PML IV was transfected into PML^{-/-} cells. If it was not transfected into PML^{-/-} cells why did the wild type PML not recruit the SUMO modified proteasome? In these types of experiments it is critical to quantitatively assess the amount of colocalisation in each cell and count at least 100 cells.

Figure 5 panels (b) and (d) have been updated to show the colocalization of the 20S proteasome and PML or PML-SIM in PML^{-/-} MEF cells to eliminate background. This Figure is much more convincing. We included the same experiments in the HEK293 cells in the supplemental section as Supplemental Figures 13 and 14. Both experiments statistically show that when cells are treated with MG132, the proteasome colocalized with PML NB only when there is a SIM present on PML. We added the following text on p. 12 of the revised manuscript:

To investigate if the SUMO interacting motif (SIM) of PML is required for the recruitment of SUMOylated proteasome to PML NBs, we transfected HEK293-SUMO3m cells with PML IV or PML IV-SIM that is mutated in its SIM core sequence (VVVI hydrophobic amino acids) (Supplementary Fig. 14). These experiments revealed that the 20S proteasome subunits co-localized with PML IV within the NBs in presence of MG132. In contrast, cells transfected with PML IV-SIM failed to recruit the proteasome under the same conditions (Supplementary Fig. 14a). To confirm these results, we performed the same experiments in PML^{-/-} MEF cells co-transfected with SUMO3 and PML IV or PML IV-SIM (Fig. 5b), which again showed that the SIM of PML was required for the recruitment of the 20S proteasome subunits to PML NBs. These results also revealed that PML IV alone, in the absence of the others isoforms, was able to recruit the proteasome upon MG132 treatment. The quantification of fluorescence intensities in both HEK293-SUMO3m and PML^{-/-} MEF cells revealed that the portion of 20S associated to PML NBs increased only in PML IV-expressing cells treated with MG132 (Supplementary Fig 14 b, Fig. 5b). Indeed, Manders' co-localization coefficient were 0.58 and 0.65 for PML IV in HEK293 SUMO3m cells (Supplementary Fig. 14b) and PML^{-/-} MEFs (Fig. 5c) treated with MG132, compared to 0.22 and 0.15 for PML IV-SIM in the same cells, respectively.

We also added the following texts to the method section:

“PML IV (accession number NP_002666.1) was used in this study. PML IV-SIM mutant was obtained by mutating the four hydrophobic residues from the SIM (VVVI) to (AAAS).”, “For quantification of 20S co-localization with PML IV or PML IV-SIM upon MG132 treatment, the JACoP plugin in ImageJ software⁵³ was used and the Manders' coefficient was calculated, n=60 cells per condition.”

3. They also observed that USP22 and USP37 were modified by SUMO and ubiquitin, and suggested that these modifications could regulate their deubiquitinase activities. There is no direct evidence for this statement, simply correlations. What would be required here is an in vitro experiment that measures the deubiquitinating activity of the protease when it is SUMO modified on the indicated site.

The reviewer is correct in stating that we do not have direct evidence to make the above statement. However, our statement referred to an observation whereby the change in activity of the DUB was correlated with the increase ubiquitylation of its substrate. This was highlighted as an observation rather than a mechanistic claim of enzymatic activity. The reviewer states that to support our observations we should perform a deubiquitylation assay of H2B that is Ubiquitylated at K121 by USP22 or USP22 Ubiquitylated at K417 and compare the results. A deep analysis of the literature highlights that site selective Ubiquitylation of substrates is still not broadly achievable, especially with natural linkages as explained in a review from 2014 (Faggiano, S. & Pastore, A. The challenge of producing ubiquitinated proteins for structural studies. Cells 3, 639-656, 2014). This means that one cannot produce H2B that is Ubiquitylated at K121 nor USP22 Ubiquitylated at K417 to perform these assays. In theory, the only mode to enzymatically produce an Ubiquitylated substrate requires previous knowledge of the E1, E2 and E3 responsible for the reaction. Moreover, USP22 must be assembled in the SAGA complex to mediate the deubiquitylated of H2B at K121. This entails that (1) we produce USP22 SUMOylated specifically at K417 (and no other Lys residues), (2) we purify the SAGA complex, (3) We reconstitute the SAGA complex with USP22 ubiquitylated at K417 (4) we produce and purify H2B Ubiquitylated at K121 and (4) we

perform the deconjugation assay of Ubiquitin at K121 of H2B in vitro. With the current technologies available, this is not achievable.

We changed the following text to down play our inference about the SUMO and ubiquitin regulation on USP22 as follows:

*On p.2 “and highlighted the unexpected regulation of deubiquitinase enzymes by ubiquitin-like modifiers and the SUMOylation of proteasome subunits” was changed to “**and highlighted co-regulation of SUMOylation and Ubiquitylation levels on deubiquitinase enzymes and the SUMOylation of proteasome subunits**”*

*On p.4 “including the regulation of deubiquitinase enzymes by UBL modifiers and the SUMOylation of the proteasome for its recruitment to PML nuclear bodies” was changed to “**including the co-regulation of SUMOylation and Ubiquitylation levels on deubiquitinase enzymes and the SUMOylation of the proteasome for its recruitment to PML nuclear bodies**”*

On p.10 we removed “We surmised that deubiquitylation of USP22 K417 could increase its activity, leading to the deubiquitylation of H2B K121 and its subsequent SUMOylation.”

*On p.13 “Interestingly, we observed that USP22 and USP37 can be modified by SUMO and ubiquitin, and that these modifications could regulate their deubiquitinase activities” was changed to “**Moreover, we observed that USP22 and USP37 can be modified by SUMO and ubiquitin at various sites simultaneously and are regulated by MG132, though their functions remain to be identified.**”*

4. The authors claim to have developed a high sensitivity MS method on the Orbitrap Fusion, If this is the case they should, using exactly the same sample, compare this new method with their previous to assess how much better it is.

The reviewer raises another valid concern. Prior to implementing the sensitive MS method for all our SUMO proteomics studies we had already validated this method with biological samples. We could not compare our current method with our previously published one since the experiments were conducted on different MS instruments. Alternatively, we compared our sensitive method to a classic/regular method used in proteomics, similarly to the comparison performed by Dr. Hay’s group in their 2014 article (PMID 24782567). We addressed this concern on p.5 as follows:

“We further analyzed the benefits of this sensitive method with biological samples where we obtained 375±24 sites for the developed method (AGC of 5e3 and a 3000 ms injection time) while garnering only 110±15 site using the classic method (AGC of 1e5 and a 50 ms injection time) for the MS/MS scans.”

5. The authors evaluated the reproducibility using "three biological replicates with three technical replicates each". However they need to clarify exactly what they mean by technical and biological replicates. Are technical replicates the same peptide prep injected into the mass spec three times or is it the same set of cells divided into 3 and carried through the workflow separately?

We clarified the terms in the text with the following addition on p. 7:

We evaluated the reproducibility using three biological replicates originating from different cell cultures with three technical replicates each, where protein extracts were separated and the workflows conducted in parallel.

6. Precisely how the authors shortlisted sites regarded as 'true' SUMOylation sites is unclear. What were the MaxQuant settings for FDR filtering? Did they just take the MaxQuant modified peptides output or were further levels of filtering applied? It is important to ascertain how confident the authors were in their site identifications. This is particularly important as many people in the SUMO field are not convinced SUMO4 is actually expressed. My assessment of the data on SUMO4 is that it is a pseudogene. However taking the data here at face value would suggest that SUMO4 is indeed expressed. Thus it is important that the identifications of modifications to SUMO4 reported here are put beyond reasonable doubt. Thus annotated spectra of the SUMOylation and ubiquitination sites of SUMO4 should be provided. This is especially important given that SUMO4 differs from other SUMOs by only a few amino-acids and many of the reported SUMO-4 peptides have the same Andromeda score as peptides from SUMO-2 and/or SUMO-3 implying they share the same identifying spectra, and so may be false identifications. This is clear from the supplied Excel data files where SUMO4 ids are often grouped into the same protein groups as SUMO2/SUMO3 IDs at conserved lysines.

We would like to thank the reviewer for this insightful comment. We analyzed all our MS/MS spectra for the SUMOylated SUMO4 peptides and realized that none of them could be directly attributed to SUMO4 beyond the shadow of a doubt due to common sequences between different SUMO paralogues. For instance the peptide with SUMO at K33 of SUMO4 shared 100% sequence with SUMO2 and are not distinguishable. Moreover, SUMO at K21 of SUMO4 could also be SUMOylation of SUMO2 assuming a single deamidation of the peptide making the two isoforms indistinguishable. Lastly, SUMO at K5 and K11 of SUMO4 could also be a SUMO2 modification where the peptide is acetylated at the N-term. For this reason we have removed all references to SUMO4 SUMOylation in the text and from Figure S9 (formally Figure S8).

We are confident in our SUMO site assignment despite the SUMO4 anomaly observed above. Indeed the sites identified were real but the protein to which they were attributed were debatable. We used a 1% FDR at the site level and used the data reported in the SUMO3.txt file created by MaxQuant for analysis. We filtered the data further where SUMO sites with a localization of <0.75 were discarded. We addressed this in the text by adding the following information on p.25:

“The false discovery rate for peptide, protein, and site identification was 1%. SUMO sites with a localization probability of > 0.75 were retained.”

However phosphorylated SUMO peptides were obtained using the evidence file and therefore, the FDR was applied at the PSM level. In order to homogenize the results presented in this manuscript, we filtered our list of Phosphorylated SUMO peptides obtained from the evidence file and only kept peptide that presented both modified and that were also identified with a 1% FDR at the site level. That list can be

found in the new Supplementary Table 2. We also changed Figure 2 panel c and d accordingly to that new table. We changed the text on p.8:

We also identified 125 SUMOylated peptides containing phosphorylated residues (localization confidence >0.75) on serine (81 sites), threonine (9 sites) (Fig. 2c, Supplementary Table 2). Phosphorylated Ser residues were primarily located 5 amino acids downstream of the SUMOylated residue, in agreement with previously published observations on phosphorylation dependent SUMOylation motif ⁸. In contrast, no correlation with respect to the SUMOylated Lys was found for phosphorylated Thr residues.

7. The authors purified SUMO modified proteins on the basis of the 6His tag on the expressed mutant version of SUMO and used this material to enrich for SUMO sites and ubiquitin sites. However more than half of the ubiquitinated proteins identified were not shown to be SUMO modified. The authors hypothesize that those unique proteins are SUMOylated on the polyubiquitin motif. This may be the case but it is also possible that this will arise from non-specific Ni-NTA binding proteins which make up the majority of Ni-NTA elutions. This would be an interesting point to sort out and it could be done by simply treating the eluted material with a recombinant SENP and analyzing the ubiquitinated proteins by Western blotting.

This comment is indeed important to address in order to ensure that the ubiquitin sites we observed in our SUMO preparation are valid. We have addressed this in two different ways. The first method employed the SENP as suggested by the reviewer. We treated cells stably expressing SUMO3 mutant (SUMO3m) cells with MG132, heat shock and As₂O₃ and subjected the Ni-NTA enriched material with SENP1 and SENP2 in combination. We found that for both MG132 and Heat shock treated cells we captured more ubiquitin than for the untreated cells and that upon SENP treatment the Ubiquitin signal vastly decreased in mass on the blot. This results shows that the ubiquitylated proteins were indeed also SUMOylated. If the proteins were not SUMOylated then the shift in mass should not have been observed upon SENP treatment. Moreover, we show that this increase in ubiquitin and SUMOylation cross-talk is not unique to MG132 treatment but also to heat shock treatments.

The second way by which we showed that the ubiquitylation resided on SUMOylated proteins was by performing western blots on Ni-NTA enriched extracts with SUMO3m cells and HEK293 wild-type cells. Upon probing for ubiquitylation we observed an increase in ubiquitin signal in the Ni-NTA enriched extract coming from SUMO3m cells upon prolonged exposure to MG132, while observing no detectable levels of ubiquitylation from Ni-NTA enriched extracts from wild-type HEK293 cells. We have appended these figures in the supplementary material as Figure S8 panels b-c and added the following text on p. 9:

However, Ni-NTA enrichment of SUMOylated proteins showed a significant enrichment of both SUMOylated and ubiquitylated proteins with a high degree of specificity whereas HEK293 cells not expressing the SUMO3m did not show the corresponding enrichment (Supplementary Fig. 8).

8. It is suggested that "SUMOylation of ribosomal subunits may prevent the formation of the protein complex, thus altering protein synthesis". If SUMO is to block something then the biological significance depends on stoichiometry. If only a small proportion of the protein is SUMO modified, then only a small proportion will be blocked from complex assembly. The authors need to consider/counter this.

Once again the reviewer has brought about an excellent point. One of the major issues in the field of PTM proteomics and even more so to SUMOylation specifically is the idea of stoichiometry of the modification. Indeed, currently little is known about the global stoichiometry of protein SUMOylation but the current belief is that it is situated below 5%. This entails that ribosomal subunits are probably SUMOylated in low stoichiometric amounts and that the changes in SUMOylation that are observed may be unfathomably small in terms of absolute quantity. We cannot rule out that the stoichiometry of SUMOylation of the ribosomal subunits may be truly small, but we must also consider that multiple SUMO sites (74 sites) on several subunits (40 subunits) are regulated by MG132 which may compound to cause a noticeable effect on the regular homeostasis of the cell. The sentence "This suggests that SUMOylation of ribosomal subunits may prevent the formation of the protein complex, thus altering protein synthesis" was changed on p.11 to:

Since SUMOylation is believed to generally occur at a low stoichiometry (<5%), the effect of a single SUMOylation event on ribosomal activity may be negligible³⁷. However, our kinetic studies have shown that 76 SUMO sites on 36 subunits are regulated by MG132. The compounded effect of several SUMOylation events occurring simultaneously may alter protein synthesis.

Reviewer #2:

1. As general feedback, many of the figure labels are too small and/or illegible. This is the case for axis labels throughout Figs 2-5. It is particularly difficult to see the image labels in Fig 5D-E. Please improve the readability of figure legends throughout the manuscript and supplement.

We would like to thank the reviewer for this comment. Some of our figures were indeed hard to read and understand due to the small fonts used in some instances. The appropriate changes have been performed on all figures to ensure clarity and readability. Fonts have been increased and images resized accordingly.

2. Proteasome inhibitors like MG132 influence many aspects of cellular signaling. It wasn't clear from the background and results text why the authors chose to perform the comprehensive proteomics analysis of Sumo-Ub crosstalk on MG132 treated cells? Was this to enhance sensitivity for the analytical method by increasing SUMO-modifications in cells? Or in contrast, was it to reveal the nature of the

underlying biology between SUMO, Ub and the proteasome. It seemed like it was both, but the experimental design and certain data interpretation are predicated on this rationale. Some additional context would be helpful in framing this study.

We agree that the explanation for using MG132 as a treatment was not clear in the text. To address this we added the following text on p.6:

We benchmarked our approach by profiling the changes in protein SUMOylation upon treatment of HEK293 cells with 10 μ M MG132, a proteasome inhibitor known to affect the levels of protein ubiquitylation and SUMOylation. We evaluated the effect of MG132 over an incubation period of 8 h to further compare the improvement of the present protocol with that of our previous study¹⁷.

3. Given that MG132 treatment depletes free ubiquitin in the cell, it is possible that SUMO is being conjugated to substrates and incorporated into ubiquitin-chains to serve as an alternate degradation signal. Work from Kim et.al. Mol Cell showed precedent for this with Nedd8 under conditions of proteasome inhibition. As such, the nature of SUMO-Ub crosstalk is highly dependent on the state of the Ub pool. That said, this topic was not discussed or considered as part of the experimental design or discussed when describing the results. Moreover, the blots in Fig. S7 do not depict 8Kd free Ub, which is critical for interpreting the decrease in signal at the 8 hr time point in the middle panel. Worth noting that historically, special methods are sometimes required to obtain good signal for mono-Ub, as most anti-Ub reagents detect polyUb better than the mono-Ub species.

The concept of interchanging Ubiquitin for other ubiquitin like modifiers under different stress environment for adaptive purposes eluded us. We thank the reviewer for mentioning this point as this could be pertinent to the discussion. We addressed this point and looked at free ubiquitin upon MG132 treatment and appended the results to Supplementary Fig. 7. Upon prolonged exposure to MG132 (up to 8 h) the ubiquitin pool is not depleted nor is the SUMO pool. These results suggest that the SUMO-Ubiquitin chains that are produced during the 8 h MG132 treatment are not caused by an alternative pathway produced by the depletion of ubiquitin. We added the following sentence on p.9:

Western blot analysis of the total cell extract revealed an overall increase of both SUMOylation and ubiquitylation levels without depleting the pool of free SUMO or ubiquitin (Supplementary Fig. 7).

4. On a related note...are the cells still alive at the 8hr timepoint after 10 μ M MG132? This is a significant exposure that in past experience may be expected to cause cells to round up and come up off the cell culture dishes. What is the nature of the His-SUMO3 expressing cells at the various time points characterized. Does expression of His-SUMO3 construct alter cell sensitivity to the time or concentration of proteasome inhibitors required to cause cell death?

This is a valid point that we should have addressed since cell death could alter the biological meaning of the data. We conduct MG132 treatments for 8 h quite often and never observed cell debris or detachment. We performed an MTT assay to obtain a quantitative assessment of cell survival for both

HEK293 and HEK293-SUMO3m cells. The results are under this comment. Both cell types respond well to the treatment and are still viable for the length of the experiment. HEK293-wt and HEK293-SUMO3m cells were untreated or treated with 10 μ M MG132 for different times. The number of viable cells was estimated using an MTT assay. Results are presented as the percentage of cells in MG132-treated cells compared to untreated cells that was arbitrarily set to 100%. We added the following sentence on p. 6:

It is noteworthy that cell viability for either HEK293 or HEK293-SUMO3m cells remained unaffected over the course of these experiments.

5. The term "crosslinking" is used to describe the method of covalently binding the antibody to the resin and preventing its elution at the end of the immunoaffinity enrichment step. Yet within the text, the authors also use this phrase to mean something different, referring to heterogeneous chains comprised of mixtures of ubiquitin and SUMO. The text should more clearly differentiate between these two and not use a single phrase interchangeably.

We have clarified the terms by using the term cross-linking exclusively for the covalent interaction of the antibody and the protein A/G beads. We used the term branched peptides rather than cross-linked when referring to SUMOylation and Ubiquitylation.

6. The authors surmise that changes in the SUMO sequence motif preference is explained by promiscuous sumoylation by Ubc9 and/or altered SUMO-E3 ligase activity following through MG132 treatment. Are the authors concerned that the motif preference could derive from a bias introduced by the mutant form of SUMO? Could it be that the introduced mutations, which occur in close proximity to the substrate motif, could be altering the conclusions drawn from these data? At the least, some discussion is required on this point.

This has been a point of concern for SUMO proteomics since all article to date use an altered form of SUMO that is already in the mature state (SENP cleaved), harbour a His tag at the N-term and often more mutations at the C-term. There are currently no methods that would allow us to determine that the “promiscuity” that we attribute to prolonged exposure to MG132 is not influenced on the mutant SUMO. We therefore acknowledge the reviews comment and included the following sentence on p.10:

We surmised that changes in the proportion of sequence motifs over time is partly explained by promiscuous SUMOylation, and/or the activation of SUMO E3 ligases brought upon by extended exposure to MG132, though we cannot exclude a possible influence attributed to the use of SUMO3 mutation on the distribution of SUMO motifs for extended periods of MG132 treatment.

7. Are the authors concerned that the antibody used to IP SUMO-remnant peptides could preferentially bind peptides with certain sub-sequences? Has the binding specificity of this 'motif-specific' antibody been thoroughly tested?

The binding specificity of the anti-body has been tested in our 2014 article with synthetic peptides where the antibody did not display a bias towards a specific backbone motif. Moreover, Figure S6 shows that 42% of the identified SUMO sites have already been identified in the literature which suggests that the mutant SUMO we used does not show a considerable bias towards a specific motif.

8. A more thorough accounting of the statistical tests should be provided in the methods section or supplement. What do the author mean when stating "reproducible label-free quantification was obtained for all biological replicates with a Pearson coefficient higher than 0.9". The numbers/method underlying this statement should be more clearly presented. Similarly, saying that "data revealed a sizeable amount cross-linked peptide" feels like an attempt to make a quantitative statement. A well defined comparison would be more appropriate.

We added the following text to the methods and materials section on p.25-26 to further explain the data analysis:

The false discovery rate for peptide, protein, and site identification was 1%. SUMO sites with a localization probability of > 0.75 were retained.

Pearson coefficients for biological reproducibility were obtained by plotting the fold changes of MG132/control for the three biological replicates.

We changed “data revealed a sizeable amount cross-linked peptide” to “data revealed more than 12 branched peptides” on p. 9 to give a quantitative value so that the reader will not have to go to the supplemental section to obtain an idea of the branching levels of SUMO and Ubiquitin.

9. Figure 5e is not referenced in the figure text...looks like a is numbering/ duplication of Fig 5D on p11.

The reviewer was correct. Indeed the second reference to Figure 5D should have read Figure 5e. The figure has been updated in the newer version of the manuscript and the oversight has been rectified.

10. Fig 5 is critical biology that is presented as the main novel element of the paper. Indeed, recruitment of proteasomes to the nuclear body via PML under conditions of stress seems potentially meaningful. For that reason, it is not sufficient to provide single cell, representative images to support these foundational claims. Systematically counted image data with sufficient n to support statistical testing would typically be required to support a claim of this sort.

Please refer to Reviewer 1's point number 2.

11. On p11 when discussing proteasome subunits, the text mistakenly notes "SUMOylation of the ribosome".

We thank the reviewer for this comment. We modified the text accordingly and added the following sentence for clarity on p. 11:

The distribution of modified residues on the outer surface of the proteasome subunits possibly implies that SUMOylation may affect protein-protein interactions.

12. Along with monoUb discussed above, histone 2A-Ub is an abundant storage depot for Ub that is also depleted upon prolonged proteasome inhibition. It appears that the blots in S7 are showing this, but it should be made more clear by capturing an extended exposure for the smaller molecular weight region of the gel. This becomes more important given that USP22, a protein highlighted as a critical crosstalk node between SUMO and Ub, is part of the SAGA complex that removed Ub from H2A. If the authors could expand upon this connection between USP22 SUMOylation, deubiquitination of H2A, the ubiquitin pools in relation to proteasome inhibitor sensitivity, this would be a meaningful biology result.

Please refer to Reviewer 1's point number 3.

13. Figure 5e is not referenced in the figure text

Please see point 9.

14. 'Lysines' (or any amino acid) should be referred to as 'lysine residues'

All references to amino acids have been corrected as per the reviewers comment.

15. Some phrases in the text are unclear and the manuscript would be easier to read and understand if these sentences were edited: for example;

o Page 4 "several unsuspected crosstalks"

Changed to "several modes of co-regulation".

o Page 6 "we also determined the number of identification obtained with these two strategies,"

Changed to “SUMOylated peptides identified by MS when using the in-solution and the cross-linked immunopurification methods”.

Reviewer #3:

1. The authors state that they enrich ubiquitylated and sumoylated peptides from sumoylated proteins. This statement is concerning. How can the authors be certain that the ubiquitin sites identified come from sumoylated proteins and not background, non-sumoylated proteins that came down in the Ni-NTA enrichment.

See response to reviewer 1 point 7.

2. What is the specificity of the Ni-NTA protein level enrichment? How would one measure that?

See response to reviewer 1 point 7, more specifically Supplemental Figure 8 panel a.

3. Why is sumo protein level enrichment needed? Is the anti-sumo remnant antibody not specific enough without an initial enrichment step? If so, this makes the workflow more tedious and prone to sample loss.

The extent of protein SUMOylation is significantly lower than protein ubiquitylation. Most of previous large scale studies (e.g. Science Signaling 7, rs2 (2014); Nature Structural & Molecular Biology 21, 927-936 (2014)) have used His-tagged SUMO mutants to enrich SUMOylated proteins. The Ni-NTA enrichment of His-SUMO s required to reduce sample complexity and enrich the proportion of SUMOylated proteins prior to the enrichment of SUMOylated peptides. Following the identification of roughly 10K SUMOylation sites, we did try to perform an IP on the total cell extract. As a result, we only identified 14 SUMOylated peptides among 18619 peptides (FDR 1%) for a total of 7 confidently identified SUMO site (localization >0.75).

4. How many peptides harbored both KGG and sumo sites?

We have added a new supplementary table (Supplementary Table 3) that contains a list of all Phosphorylated SUMO peptides and Ubiquitylated SUMO peptides. We identified a total of 63 Ubiquitylated SUMO peptides corresponding to 58 ubiquitylation sites including 44 peptides, 43 of these sites stemming from the 2DLC experiment.

5. Why is KGG enrichment completed first? There is no explanation for why this workflow was chosen. *The occurrence of ubiquitylated peptides following Ni-NTA purification is expected to be much lower than SUMOylated peptides and immunoaffinity purification of the Gly-Gly containing tryptic peptides was performed first to minimize sample losses. We added the following sentence on p. 9 to clarify:*

Since the occurrence of ubiquitylated peptides following Ni-NTA purification is expected to be much lower than SUMOylated peptides, we first performed immunoaffinity purification of the Gly-Gly containing tryptic peptides to minimize sample losses.

6. Why is the yield of sites from the sequential sumo and ubiquitin experiments low (1620 sumo sites single time point and 778 in consecutive time points) compared to the initial experiments described in the paper (~10K sites)? Was the optimized workflow not used?

All kinetic experiment were performed using 4 mg of Ni-NTA purified protein extracts corresponding to lower amounts than that used for comprehensive SUMO proteome analyses. Also, reverse phase LC-MS/MS and not 2D-LC-MS/MS was performed to facilitate the comparison of datasets since all experiments used label-free quantification. Finally, we only selected SUMO peptides that were identified in at least 3 consecutive time points thus reducing the number of SUMO peptides for which temporal profiles were obtained.

7. If cells were not pre-treated with MG-132 to increase sumo stoichiometry, would this workflow still enable enrichment of >1000 or >5000 sumo sites? What protein input amount would be needed?

The comparison of SUMO sites identified with and without MG132 is shown in Supplementary Fig. 5. For example a total of 877 SUMO peptides were identified from 4 mg of Ni-NTA purified proteins compared to 1527 SUMO peptides when cells were treated with MG132.

Reviewer Comments:

Reviewer #1 (Remarks to the Author):

I think the authors have addressed most of the points that I raised in an adequate fashion and the paper should now be suitable for publication.

Reviewer #2 (Remarks to the Author):

The revised manuscript entitled “Uncovering the SUMOylation and Ubiquitylation Crosstalks in Human Cells Using Sequential Peptide Immunopurification” has been substantially improved during this revision and is nearly acceptable for publication. The text is fair and effective at describing their method for detecting site specific sumoylation and its benefits for characterizing crosstalk with ubiquitin. Technical discussion and figures are clear and informative, providing interesting details involving crosslinking, automatic gain control input amounts and their effects on LC-MS sensitivity and specificity. Their total yield of >9800 sites within 6 SCX fractions is quite impressive, while their description of the data is informative and even handed. The biological follow up studies are substantially strengthened by the addition of data from the PML mutant proteins. Remaining questions and concerns are listed below:

- On p9 the text indicates that the free Ub and Sumo pools are not depleted by MG132. Comparing t=0 and t=2 in the free ubiquitin blot from Fig. S7, it appears to be a marked decrease in free ubiquitin. Is this a correct assessment and is this consistent with the authors observations in repeating these experiments. Similarly, panel S8B shows that both MG132 and heat shock lead to depletion of the free Sumo pool. This is arguably the case at 4hr for free Sumo in S7 as well.
- I did not see any text referencing the subpanels in Fig S8...it didn't appear that S8B was referenced at all. Moreover, it seemed that S7 and S8C were more or less redundant, but showed different results in the NTA beads portion of the blot. The authors should clarify matters relating to these two Supplemental figures.
- The additional data described on p12 Results and p13 Discussion about the PML mutant proteins is a valuable addition to the paper and would be better served in the main figures (i.e. Fig. S14) rather than in the Supplement.
- The text uses the term ‘crosstalks’ in the consistently throughout (including title). This term would be better replaced by the singular ‘crosstalk’.

Reviewer #3

No comments for the author.

Response to review of Manuscript "Uncovering the SUMOylation and Ubiquitylation Crosstalks in Human Cells Using Sequential Peptide Immunopurification" - **NCOMMS-16-14810A**

Italic highlights response on specific comments raised by the reviewers

Bold highlights texts that was modifications in the revised manuscript, and tabulated with track changes in that actual document

We thank once again all reviewers for their positive feedback and constructive criticisms.

Reviewer #2:

1. On p9 the text indicates that the free Ub and Sumo pools are not depleted by MG132. Comparing t=0 and t=2 in the free ubiquitin blot from Fig. S7, it appears to be a marked decrease in free ubiquitin. Is this a correct assessment and is this consistent with the authors observations in repeating these experiments. Similarly, panel S8B shows that both MG132 and heat shock lead to depletion of the free Sumo pool. This is arguably the case at 4hr for free Sumo in S7 as well.

We thank the reviewer for raising this point. It is true there is a significant decrease of both free SUMO and free ubiquitin upon MG132 treatment; however we thought the word "depletion" was meant as an absolute exhaustion. Since this word can be interpreted in a couple ways we modified the text in order to better express this observation:

Western blot analysis of the total cell extract revealed an overall increase of both SUMOylation and ubiquitylation levels coupled with a ~20% decrease in the pool of free SUMO or ubiquitin (Supplementary Fig. 7).

2. I did not see any text referencing the subpanels in Fig S8...it didn't appear that S8B was referenced at all.

We did not reference subpanels of the Supplementary Fig. 8 in the manuscript. We apologize and reorganized Supplementary Fig. 8 so that the results followed the text and added references to each panel accordingly:

However, Ni-NTA enrichment of SUMOylated proteins showed a significant enrichment of both SUMOylated and ubiquitylated proteins with a high degree of specificity in HEK293 cells expressing the SUMO3 mutant protein (Supplementary Fig. 7) whereas WT HEK293 cells did not show the corresponding enrichment (Supplementary Fig. 8a). Moreover, we confirmed that the ubiquitylation signal on our western blots did not arise only from ubiquitylation on polySUMO chains by performing an *in vitro* deSUMOylation assay (Supplementary Fig. 8b). Indeed, a portion of the ubiquitin signal on western blot appears to stem from ubiquitin that is directly attached to the target substrate. We also assessed the extent of non-specific binders arising from the NiNTA purification by comparing the number of identified SUMOylated protein in our comprehensive analysis to the raw NiNTA extract (Supplementary Fig. 8c). After NiNTA enrichment proteins were digested with trypsin and

ubiquitylated peptides were enriched using the anti-K(GG) antibody cross linked to the agarose beads to limit nonspecific binding¹.

3. Moreover, it seemed that S7 and S8C were more or less redundant, but showed different results in the NTA beads portion of the blot. The authors should clarify matters relating to these two Supplemental figures.

*We did not describe the difference between S7 and S8C appropriately in the text, which leads to confusion. Supplementary Fig. 7 was obtained using extract from the SUMO3m stable cell line expressing the His-SUMO3, while the blot on Supplementary Fig. 8 was obtained using extract from WT HEK293 cells, therefore do not express His-SUMO. To avoid this confusion, we modified the text as follow: **However, Ni-NTA enrichment of SUMOylated proteins showed a significant enrichment of both SUMOylated and ubiquitylated proteins with a high degree of specificity in HEK293 cells expressing the SUMO3 mutant protein (Supplementary Fig. 7) whereas WT HEK293 cells did not show the corresponding enrichment (Supplementary Fig. 8a).***

4. The additional data described on p12 Results and p13 Discussion about the PML mutant proteins is a valuable addition to the paper and would be better served in the main figures (i.e. Fig. S14) rather than in the Supplement.

We included the reviewer's comment and modified Figure 5 by removing panel 5b and 5c and adding a new Figure 6 that combines IFs and quantification from Supplementary Figure 14 and Figure 5. The legend and text was modified accordingly.

5. The text uses the term 'crosstalks' in the consistently throughout (including title). This term would be better replaced by the singular 'crosstalk'.

This has been corrected in the new version of the manuscript

1. Udeshi, N.D. et al. Refined preparation and use of anti-diglycine remnant (K-ε-GG) antibody enables routine quantification of 10,000s of ubiquitination sites in single proteomics experiments. *Molecular & cellular proteomics: MCP* **12**, 825-831 (2013).